# Screening and effective RPA-like charge susceptibility in the extended Hubbard model

A. Al-Eryani[1], S. Heinzelmann[2], K. Fraboulet[2,4], F. Krien[3], S. Andergassen[3,4],

**1** Theoretical Physics III, Ruhr-University Bochum, 44801 Bochum, Germany
**2** Institut für Theoretische Physik and Center for Quantum Science, Universität Tübingen, Auf der Morgenstelle 14, 72076 Tübingen, Germany
**3** Institute for Solid State Physics, Vienna University of Technology, 1040 Vienna, Austria
**4** Institute of Information Systems Engineering, Vienna University of Technology, 1040 Vienna, Austria
* aiman.al-eryani@rub.de

December 11, 2024

## Abstract

We generalize the recently introduced single-boson exchange formalism to non-local interactions. In the functional renormalization group application to the extended Hubbard model in two dimensions, we show that the flow of the rest function can be neglected up to moderate interaction strengths. We explore the physics arising from the interplay between onsite and nearest-neighbor interactions in various parameter regimes by performing a fluctuation diagnostics. Differently from the magnetic and superconducting susceptibilities, the charge susceptibility appears to be described by the random phase approximation (RPA). We show that this behavior can be traced back to cancellations in the renormalization of the density fermion-boson coupling.

# 1 Introduction

The functional renormalization group (fRG) has played an important role as an unbiased method to study strongly correlated systems[1], and in particular the Hubbard model [3] as a prototypical model for interacting fermions. The fRG is unbiased in the sense that all channels are treated on equal footing. Due to its versatility, it can be applied to arbitrary dimension, lattice geometries (including frustration), fillings, and in particular also to nonlocal interactions. However, the numerical solution of the RG flow is in general challenging and truncations are applied in practical implementations. Specifically, the expansion of the Wetterich equation in terms of $n$-particle irreducible vertex functions is typically truncated at the level of the two-particle vertex. Neglecting the renormalization of three- and higher-order particle vertices yields approximate one-loop equations for the self-energy and the two-particle vertex. This typically leads to a limitation to the weak to moderate coupling regime, where forefront algorithmic advancements brought the fRG for interacting fermions on two-dimensional (2D) lattices to a quantitatively reliable level [4,5]. Extensions incorporate contributions of the three-particle vertex [6], up to a fully two-particle self-consistent scheme in the multiloop fRG [4,5,7,8], which is formally equivalent to the parquet approximation. Strongly correlated parameter regimes are accessible by exploiting the dynamical mean-field theory (DMFT) [9,10] as a starting point for the fRG flow [11–13].

These advancements go hand in hand with the development of efficient parametrization schemes for the two-particle vertex. While naïve power counting arguments suggest the frequency dependence to be irrelevant at weak coupling, it substantially affects the results on a quantitative level [14,15]. To that end, efforts have been made toward a full frequency treatment [15] which includes the high-frequency asymptotics [16,17]. Its combination with the so-called "truncated unity" (TU) fRG [14,18,19] using the channel decomposition in conjunction with a form-factor expansion for the fermionic momentum dependence brought the fRG for interacting fermions on 2D lattices to a quantitatively reliable level in the weak to moderate coupling regime [5]. The underlying algorithmic advancements have been established for the conventional Hubbard model with an on-site Coulomb interaction and are not trivially generalized to nonlocal interactions [20,21]. We here aim to address this point, providing a numerically feasible fRG-based computation scheme for treating a nearest-neighbor interaction.

Since the long-range Coulomb interaction is often only partially screened, the effects of more extended interactions have to be taken into account. In addition to the antiferromagnetic fluctuations induced by the on-site interaction $U$, the presence of a nearest-neighbor interaction $U'$ introduces charge-density wave fluctuations [22–26]. Their competition leads to a phase transition between a spin-density wave and a charge-density

---

[1]See Refs. [1,2] for a review.

wave in the low-temperature regime. Early analytical calculations predicted the transition to occur at values of $U' \gtrsim U/4$ [27,28], which has been confirmed by quantum Monte Carlo (QMC) results [23], the Hartree-Fock approximation and the dynamical mean-field theory (DMFT) [29], the two-particle self-consistent approach (TPSC) [30, 31], the extended DMFT (EDMFT) [32,33], the dynamical cluster approximation (DCA) [24–26,34], the dual boson approach [35–37], the parquet approximation [38], the slave-boson approach [39,40], as well as the determinantal QMC (DQMC) [41,42]. Moreover, the nearest-neighbor interaction also renormalizes the local interaction, effectively screening it [43].

With the fRG, a previous analysis at small to moderate doping showed that antiferromagnetic spin fluctuations in the pairing and charge channels generate attractive $d$-wave interactions [44]. The resulting nematic fluctuations are enhanced by the nearest-neighbor interaction in the extended Hubbard model, even though they remain always smaller than the dominant antiferromagnetic, pairing, or charge-density wave fluctuations. More recently, it has been shown that a single-band Hubbard model with extended interactions on a triangular lattice successfully captures the physics of moiré hetero-bilayers of transition metal dichalcogenides [45]. A static fRG study of this model [46] has found a phase diagram with a large variety of exotic forms of superconductivity [47–52].

Recently, the single-boson exchange (SBE) decomposition [53] has shown to provide a physically intuitive and also computationally efficient description of the relevant fluctuations in terms of single boson exchanges. At weak coupling, the multiboson processes are irrelevant [13,54]. They become important only in the vicinity of the pseudo-critical transition observed in the one-loop approximation [54]. At strong coupling, the advantages of the SBE formalism are of particular relevance. In this regime, the multiple divergences in the two-particle irreducible vertex functions [55–69] make the applicability of conventional Bethe-Salpeter equations and/or parquet formalism [16,70] rather problematic. In contrast, no diagrammatic element of the SBE decomposition of the vertex function displays [53,71] the non-perturbative divergencies that plague their parquet counterparts. Since only the multiboson processes depend on three independent frequency and momentum variables, neglecting them drastically reduces the computational complexity of the problem. In contrast, the effective bosonic interaction can be represented by bosonic propagators and fermion-boson couplings (or Hedin vertices) [72] determined from the vertex asymptotics, in analogy to the kernel functions encoding the high-frequency asymptotics [17]. They depend on one and two independent arguments, respectively, and therefore their numerical treatment including the full frequency and momentum dependence is much less demanding. We here present a generalisation of the SBE decomposition to nonlocal interactions. In particular, we will provide a numerically feasible fRG scheme to explore the rich physics of the extended Hubbard model in the different parameter regimes. The most remarkable finding is the linear increase of the density or charge susceptibility with $U'$. At the same time, the fermion-boson coupling in the same channel appears to be independent on the nearest-neighbor interaction. As emerges from the fluctuation diagnostics, this observation is at the origin of an effective RPA-like behavior resulting from cancellations in the renormalization of the fermion-boson couplings.

The paper is organized as follows: In Section 2, we introduce the extended Hubbard model. We briefly review the SBE formulation of the fRG in Section 3, before presenting its extension to the treatment of extended interactions. We then show how this decomposition is implemented in the fRG and provide the theoretical background underlying the fluctuation diagnostics analysis. In Section 4, we first present a detailed analysis of different approximations that allow us to considerably reduce the numerical effort of the algorithmic implementation. We then investigate the influence of a nonlocal interaction on the physics at half filling and finite doping. We conclude with a summary in Section 5.

## 2   Extended Hubbard model

The Hamiltonian for the extended Hubbard model in 2D reads

$$H = -t \sum_{<ij>\sigma} c_{i\sigma}^\dagger c_{j\sigma} - t' \sum_{\ll ij \gg \sigma} c_{i\sigma}^\dagger c_{j\sigma} + \frac{U}{2} \sum_{i\sigma} n_{i\sigma} n_{i-\sigma} + \frac{U'}{2} \sum_{<ij>\sigma\sigma'} n_{i\sigma} n_{j\sigma'} - \mu \sum_{i\sigma} n_{i\sigma}, \quad (1)$$

where $c_{i\sigma}$ $(c_{i\sigma}^\dagger)$ annihilates (creates) an electron with spin $\sigma$ at the lattice site $i$ $(n_{i\sigma} = c_{i\sigma}^\dagger c_{i\sigma})$, $t_{ij} = -t$ is the hopping between neighboring and $t_{ij} = -t'$ between next-nearest neighboring sites on a square lattice, $\mu$ the chemical potential (with half filling corresponding to $\mu = 0$). $U$ is the local on-site Hubbard interaction and $U'$ the nearest-neighbor interaction, which accounts, for example, for longer-ranged matrix elements of the Coulomb repulsion. In momentum space, we find (omitting the momentum space volume factors)

$$H = \sum_{\mathbf{k}\sigma} (\epsilon(\mathbf{k}) - \mu) c_{\mathbf{k}\sigma}^\dagger c_{\mathbf{k}\sigma} + \frac{1}{2} \sum_{\sigma\sigma'} \sum_{\mathbf{k}\mathbf{k}'\mathbf{q}} V_0(\mathbf{q}) c_{\mathbf{k}\sigma}^\dagger c_{\mathbf{k}+\mathbf{q}\sigma} c_{\mathbf{k}'+\mathbf{q}\sigma'}^\dagger c_{\mathbf{k}'\sigma'}. \quad (2)$$

For the square lattice, the free dispersion is given by

$$\epsilon(\mathbf{k}) = -2t(\cos k_x + \cos k_y) - 4t' \cos k_x \cos k_y. \quad (3)$$

The bare propagator is then given by

$$G_0(\mathbf{k}, i\nu) = \left( i\nu + \mu - \epsilon(\mathbf{k}) \right)^{-1}, \quad (4)$$

while the bare interaction reads

$$V_0(\mathbf{q}) = U + 2U'(\cos q_x + \cos q_y). \quad (5)$$

In the following, we use $t \equiv 1$ as the energy unit.

## 3   Methods

### 3.1   Single-boson exchange formulation

The basic idea underlying the SBE formalism, is to express the two-particle vertex in terms of different flavors of collective excitations, represented by effective exchange bosons. Besides the physically intuitive picture, this provides a key to identify instabilities.

Recently, an alternative to the two-particle reducibility has been introduced for local interactions: the notion of bare interaction or $U$-reducibility [53] featuring the more pronounced vertex dependence on bosonic arguments. Diagrams that can be split into two separate parts by cutting a bare interaction are termed $U$-reducible and represent the exchange of a single boson. Diagrams that can not, are $U$-irreducible and correspond to multiboson processes. As for the two-particle reducibility, these $U$-reducible diagrams can be classified into diagrammatic channels (these are best suited for the discussion of diagrammatic properties, for the translation to physical channels, see Appendix A): particle-hole ($ph$), particle-hole crossed ($\overline{ph}$), and particle-particle ($pp$). All $U$-reducible diagrams are therefore also two-particle reducible, with the exception of the bare interaction, which is considered to be $U$-reducible in all three channels. The reverse is however not true (two-particle reducible can be $U$-irreducible). The sum of all $U$-reducible diagrams in a given diagrammatic channel $r = ph$, $\overline{ph}$, $pp$, including the bare interaction, is

Figure 1: Diagrammatic representation of the parquet equation in the SBE formulation, in the parquet approximation.

denoted by $\nabla_r$ and represents the exchange of a single bosonic propagator $w_r$ between two fermion-boson couplings $\lambda_r$

$$\nabla_r(q, k, k') = \lambda_r(q, k) w_r(q) \bar{\lambda}_r(q, k') \tag{6}$$

for $V_0^r = U$. All diagrams with multiboson exchange processes are contained in the rest function $M_r(q, k, k')$. We note that $\bar{\lambda}_r = \lambda_r$ in the present notation [13], therefore we restrict ourselves to $\lambda_r$ in the following. The bosonic propagator $w_r$ represents the exchange of a single boson from a diagrammatic point of view and is closely related to the $s$-wave susceptibility by

$$\chi_r(q) = \frac{V_0^r - w_r(q)}{(V_0^r)^2} \tag{7}$$

recalling the original idea of putting collective excitations center stage. $V_0^r = U$ is the bare two-particle vertex expressed in channel $r$ and corresponds simply to the on-site interaction $U$ for the Hubbard model (with $U' = 0$). The parquet decomposition [70] is obtained by expressing $\Phi_r = \nabla_r - V_0^r + M_r$ in the SBE formulation

$$V^r(q, k, k') = \nabla_r(q, k, k') + M_r(q, k, k') + \sum_{r' \neq r} P^{r' \to r} (\nabla_{r'} + M_{r'})(q, k, k') + V_{\text{DC}}^r + V_{\text{2PI}}^r, \tag{8}$$

where $V_{\text{2PI}}$ is the two-particle irreducible part of the vertex, $V_{\text{DC}}^r = -3V_0^r$ accounts for the double counting of the bare vertex for each $\nabla^r$ and $P^{r' \to r}$ translates between the respective channel parametrizations. In the parquet approximation, $V_{\text{2PI}}^r = V_0^r = U$, see Fig. 1.

The advantage of this scheme for the algorithmic implementation is evident from Eq. (6): the quantity $\nabla_r$ that depends on three independent indices can be factorized into quantities that depend on two arguments at most.

We also note that the SBE representation largely simplifies treating the frequency dependence in fRG implementations, as compared to the schemes based on the parquet decomposition, since no additional flow equations are required. Instead, due to the high-frequency limits

$$\lim_{\Omega \to \infty} w_r(\Omega, \mathbf{q}) = V_0^r \tag{9a}$$

$$\lim_{\Omega / \nu \to \infty} \lambda_r(\Omega, \mathbf{q}, \nu, \mathbf{k}) = 1 \tag{9b}$$

$$\lim_{\Omega / \nu / \nu' \to \infty} M_r(\Omega, \mathbf{q}, \nu, \mathbf{k}, \nu', \mathbf{k}') = 0 \tag{9c}$$

that coincide with the initial values for the flow, the high-frequency behavior of the vertex is automatically included as

$$\lim_{\Omega \to \infty} V^r(\Omega, \mathbf{q}, \nu, \mathbf{k}, \nu', \mathbf{k}') = V_0^r \tag{10a}$$

$$\lim_{\nu \to \infty} V^r(\Omega, \mathbf{q}, \nu, \mathbf{k}, \nu', \mathbf{k}') = w_r(\Omega, \mathbf{q}) \lambda_r(\Omega, \mathbf{q}, \nu', \mathbf{k}') \tag{10b}$$

$$\lim_{\nu' \to \infty} V^r(\Omega, \mathbf{q}, \nu, \mathbf{k}, \nu', \mathbf{k}') = \lambda_r(\Omega, \mathbf{q}, \nu, \mathbf{k}) w_r(\Omega, \mathbf{q}) \tag{10c}$$

$$\lim_{\nu/\nu'\to\infty} V^r(\Omega, \mathbf{q}, \nu, \mathbf{k}, \nu', \mathbf{k}') = w_r(\Omega, \mathbf{q}). \tag{10d}$$

The SBE formalism has been introduced for local interactions and applied exclusively to these up to now. In the following, we will extend it to nonlocal interactions.

## 3.2  Extension to nonlocal interactions

The momentum dependence introduced by the nonlocal interaction in the extended Hubbard model has to be accounted for in a way that captures the relevant physics without dramatically increasing the numerical effort for its treatment. The SBE formalism, as established for local bare interactions, relies on the classification of diagrams according to their reducibility in $U$. With an additional $U'$, the exact form of the bare vertex depends on the channel

$$V_0^{ph}(\mathbf{q}, \mathbf{k}, \mathbf{k}') = U + 2U'\left(\cos(q_x) + \cos(q_y)\right) \tag{11a}$$

$$V_0^{\overline{ph}}(\mathbf{q}, \mathbf{k}, \mathbf{k}') = U + 2U'(\cos(k_x - k_x') + \cos(k_y - k_y')) \tag{11b}$$

$$V_0^{pp}(\mathbf{q}, \mathbf{k}, \mathbf{k}') = U + 2U'(\cos(k_x - k_x') + \cos(k_y - k_y')), \tag{11c}$$

as illustrated diagrammatically in Fig. 2. While there is only one bare interaction, its momentum dependence exhibits different forms in the different channel notations. Note that the bosonic momentum enters exclusively in the particle-hole channel, while the particle-hole crossed and particle-particle channel exhibit a fermionic momentum dependence. Translated to physical channels X = M, D, and SC (see Appendix A), the bare interaction reads [38, 73]

$$V_0^{\mathrm{M}}(\mathbf{q}, \mathbf{k}, \mathbf{k}') = -U - 2U'(\cos(k_x - k_x') + \cos(k_y - k_y')) \tag{12a}$$

$$V_0^{\mathrm{D}}(\mathbf{q}, \mathbf{k}, \mathbf{k}') = U + 4U'(\cos(q_x) + \cos(q_y)) - 2U'(\cos(k_x - k_x') + \cos(k_y - k_y')) \tag{12b}$$

$$V_0^{\mathrm{SC}}(\mathbf{q}, \mathbf{k}, \mathbf{k}') = U + 2U'(\cos(k_x - k_x') + \cos(k_y - k_y')), \tag{12c}$$

for the magnetic, density, and superconducting channel respectively. The superconducting contribution can be further split into a singlet and a triplet component

$$V_0^{\mathrm{sSC}}(\mathbf{q}, \mathbf{k}, \mathbf{k}') = V_0^{\mathrm{SC}}(\mathbf{q}, \mathbf{k}, \mathbf{k}') + V_0^{\mathrm{SC}}(\mathbf{q}, \mathbf{k}, \mathbf{q} - \mathbf{k}') \tag{13a}$$

$$V_0^{\mathrm{tSC}}(\mathbf{q}, \mathbf{k}, \mathbf{k}') = V_0^{\mathrm{SC}}(\mathbf{q}, \mathbf{k}, \mathbf{k}') - V_0^{\mathrm{SC}}(\mathbf{q}, \mathbf{k}, \mathbf{q} - \mathbf{k}'), \tag{13b}$$

where differently to the Hubbard model, the triplet contribution does not vanish. We will show how the SBE formulation can be adapted to include the nearest-neighbour interaction, see also Ref. [70, 74, 75]. By redefining the notion of bare interaction reducibility, we will be able to retain the advantages of both a direct physical interpretation and a reduced numerical effort. This is however not achieved by a straightforward extension, which turns out to be very inefficient, see Appendix B.

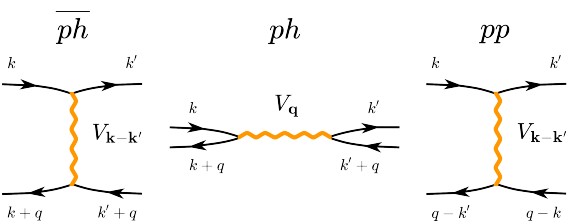

Figure 2: Bare interaction of the extended Hubbard model in the different channels.

In order to extend the SBE to a nonlocal bare interaction, it is helpful to revisit the original idea underlying its introduction: collective excitations. The nearest-neighbor interaction enhances the charge-density wave fluctuations fully encoded in $w^{\mathrm{D}}$. The influence of $U'$ on the other objects is expected to be of secondary importance. Can this be translated into a redefinition of the notion of bare interaction reducibility that includes the influence of $U'$ on the charge-density wave, but does not destroy the numerically favourable factorization of $\nabla^{\mathrm{X}}(q, k, k') = \lambda^{\mathrm{X}}(q, k) w^{\mathrm{X}}(q) \lambda^{\mathrm{X}}(q, k')$?

This can indeed be achieved by separating the purely bosonic part and the one depending on fermionic arguments

$$V_0^{\mathrm{X}}(\mathbf{q}, \mathbf{k}, \mathbf{k}') = \mathcal{B}^{\mathrm{X}}(\mathbf{q}) + \mathcal{F}^{\mathrm{X}}(\mathbf{k}, \mathbf{k}'). \tag{14}$$

The next step is to generalize the notion of bare interaction reducibility to $\mathcal{B}$-*reducibility*, see Fig. 3 for the diagrammatic representation. In the model we are considering, the bosonic bare interaction reads

$$\mathcal{B}^{\mathrm{D}}(\mathbf{q}) = U + 4U'(\cos(q_x) + \cos(q_y)) \tag{15}$$

for the density channel, and $-\mathcal{B}^{\mathrm{M}}(\mathbf{q}) = \mathcal{B}^{\mathrm{SC}}(\mathbf{q}) = U$ for the magnetic and superconducting one, while the fermionic bare interaction is given by

$$\mathcal{F}^{\mathrm{D}}(\mathbf{k}, \mathbf{k}') = -2U'(\cos(k_x - k_x') + \cos(k_y - k_y')), \tag{16}$$

with $\mathcal{F}^{\mathrm{M}}(\mathbf{k}, \mathbf{k}') = \mathcal{F}^{\mathrm{D}}(\mathbf{k}, \mathbf{k}') = -\mathcal{F}^{\mathrm{SC}}(\mathbf{k}, \mathbf{k}')$.

The expansion in form factors together with the projections between the channels are reported in Appendix C. This splitting guarantees that the bosonic propagator retains purely bosonic dependence. The $\mathcal{B}$-irreducible vertex is then given by

$$\mathcal{I}^{\mathrm{X}} = V^{\mathrm{X}} - \nabla^{\mathrm{X}}, \tag{17}$$

where $\nabla^{\mathrm{X}}$ now corresponds to the $\mathcal{B}$-reducible vertex, instead of the $U$-reducible vertex in channel X. In analogy to the conventional SBE, the construction of the full vertex $V$ from $\nabla^{\mathrm{X}}$ and the two-particle reducible $\mathcal{B}^{\mathrm{X}}$-irreducible vertex $M^{\mathrm{X}}$, $\mathcal{B}^{\mathrm{X}}$ occurring in each $\nabla^{\mathrm{X}}$, should be subtracted to prevent double counting. However, for a density-density interaction, all long-ranged contributions (including $U'$) cancel with those from the $V_0$ contribution to the two-particle irreducible vertex, see Appendix A. This means that the equation for the full vertex $V$ is identical to the one of the Hubbard model shown in Fig. 1, but with $\nabla^{\mathrm{X}}$ and $M^{\mathrm{X}}$ defined with respect to $\mathcal{B}$-reducibility instead of $V_0$-reducibility. Note that this notion of $\mathcal{B}$-reducibility depends on the choice of channel notation, i.e.

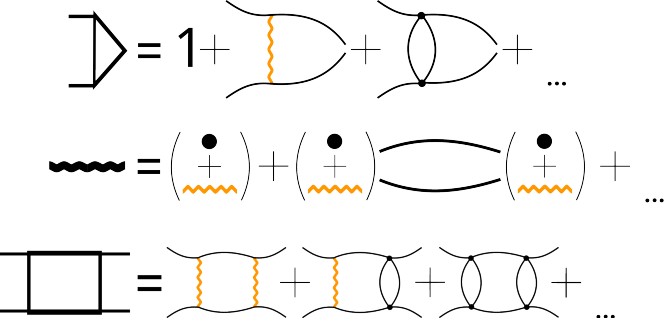

Figure 3: Diagrammatic illustration of the bare interaction reducibility adapted to the nonlocal interaction of the extended Hubbard model.

the bosonic and fermionic arguments for the bare interactions defined above can change their nature when translated between the different channel, see Appendix C. Nonetheless, under this decomposition, no additional dependencies are generated. In particular, in the natural channel notation no additional sums over form factors are introduced in $w^{\mathrm{X}}$, $\lambda^{\mathrm{X}}$ and $M^{\mathrm{X}}$. As a consequence, the multiplicative nature of $\nabla^{\mathrm{X}}$ is upheld.

On the other hand, both the fermion-boson couplings and the rest functions develop non-trivial high-frequency asymptotics that require additional computing resources. A priori, it is to be seen whether the rest function (as well as its flow) can be neglected in this formulation. For models with extended bare interactions, the momentum dependence of the full vertex $V$ diagrammatically no longer results only from propagators $G(i\nu, \mathbf{k})$ in which momentum and frequency dependence always appear together and which decay to zero for large frequencies. The extended interactions will give rise to an additional momentum dependence not tied to the frequency dependence that does not vanish in the asymptotic regime. Consequently, the high-frequency asymptotic behavior of the SBE objects will display a non-trivial dependence on the momentum and the asymptotics given in Eqs. (9b) and (9c) do not hold in general[2]. Also, the relation to the $s$-wave susceptibility (see Eq. (7)) is naturally replaced by

$$\chi^{\mathrm{X}}(\Omega, \mathbf{q}) = \frac{\mathcal{B}^{\mathrm{X}}(\mathbf{q}) - w^{\mathrm{X}}(\Omega, \mathbf{q})}{(\mathcal{B}^{\mathrm{X}}(\mathbf{q}))^2}. \tag{18}$$

So far no approximation has been made. The $U'$-dependent part of the bare interaction has been included in the irreducible vertex of the magnetic and superconducting channels and cancels out for density-density interactions. This choice is motivated by the expected effect of $U'$ on $w^{\mathrm{D}}$ that accounts for the charge-density wave tendencies. As long as the rest function and high-frequency asymptotics are included, all contributions are taken into account in an unbiased way and charge-density wave fluctuations are not favored over other possible ordering tendencies. At the same time, if the latter prevail, the present scheme allows us to neglect the rest function flow providing an efficient numerical computation scheme.

To summarize, the SBE implementation for the conventional Hubbard model can be easily extended by

1. Replacing the initial value of $w^{\mathrm{D}} = U \to U + 4U'(\cos(q_x) + \cos(q_y))$ and modifying the definition of $\mathcal{I}^{\mathrm{X}}$, where instead of subtracting $2V_0^{\mathrm{X}}$ only $2U$ is subtracted.

2. Taking care of the non-trivial high-frequency asymptotics of $\lambda^{\mathrm{X}}$ and $M^{\mathrm{X}}$.

To conclude this section, we note that the splitting of the bare interaction into bosonic and fermionic parts in Eq. (14) is analogous to Hedin's treatment of the Coulomb interaction [76], where exciton-like diagrams are absorbed in the polarization function (see, for example, Ref. [77]). Interestingly, also in Hedin's original formalism, the fermion-boson vertex could retain some residual momentum dependence in the high-frequency limit, a question which may be considered in future work. In conclusion, we here extend Hedin's theory for nonlocal interactions by including the crossing symmetry, generalizing the SBE decomposition for systems with a local Hubbard interaction [53].

---

[2]One exception is when the extended bare interaction itself decays in frequencies. In this case, the high-frequency asymptotics becomes trivial again, as for retarded interactions that are extended in time due to e.g. phonons.

## 3.3 Functional renormalization group implementation

In this section we discuss how the generalized SBE decomposition introduced above is applied in the fRG (for the comparison with the straightforward implementation of the SBE approach in terms of $V_0$-reducibility, the reader is referred to Appendix B).

Except for two differences, the flow equations are identical to those of the Hubbard model introduced previously in Refs. [54, 78]. We report them here for completeness:

$$\dot{w}^{\mathrm{X}}(\Omega, \mathbf{q}) = -\mathrm{Sgn\,X}\left(w^{\mathrm{X}}(\Omega, \mathbf{q})\right)^2 \sum_{m,m',\nu} \lambda^{\mathrm{X}}(\Omega, \mathbf{q}, \nu, m)\dot{\Pi}^{\mathrm{X}}(\Omega, \mathbf{q}, \nu, m, m')\lambda^{\mathrm{X}}(\Omega, \mathbf{q}, \nu, m')$$

(19a)

$$\dot{\lambda}^{\mathrm{X}}(\Omega, \mathbf{q}, \nu, n) = -\mathrm{Sgn\,X} \sum_{m,m',\nu'} \lambda^{\mathrm{X}}(\Omega, \mathbf{q}, \nu', m)\dot{\Pi}^{\mathrm{X}}(\Omega, \mathbf{q}, \nu', m, m')\mathcal{I}^{\mathrm{X}}(\Omega, \mathbf{q}, \nu', m', \nu, n)$$

(19b)

$$\dot{M}^{\mathrm{X}}(\Omega, \mathbf{q}, \nu, n, \nu', n') = -\mathrm{Sgn\,X} \sum_{m,m',\nu''} \mathcal{I}^{\mathrm{X}}(\Omega, \mathbf{q}, \nu, n, \nu'', m)\dot{\Pi}^{\mathrm{X}}(\Omega, \mathbf{q}, \nu'', m, m')$$
$$\times \mathcal{I}^{\mathrm{X}}(\Omega, \mathbf{q}, \nu'', m', \nu', n')',$$

(19c)

with $\mathrm{Sgn\,SC} = -\mathrm{Sgn\,D} = -\mathrm{Sgn\,M} = 1$. The dot denotes the derivative with respect to the RG scale $\Lambda$. In particular, the derivative of the bubble is given by

$$\dot{\Pi}^{\mathrm{M/D}}(\Omega, \mathbf{q}, \nu, n, n') = -\int d\mathbf{p} f_n(\mathbf{p}) f_{n'}(\mathbf{p}) G^{\Lambda}(\nu, \mathbf{p}) S^{\Lambda}(\Omega + \nu, \mathbf{q} + \mathbf{p}) + (G \leftrightarrow S) \quad (20a)$$

$$\dot{\Pi}^{\mathrm{SC}}(\Omega, \mathbf{q}, \nu, n, n') = \int d\mathbf{p} f_n(\mathbf{p}) f_{n'}(\mathbf{p}) G^{\Lambda}(\nu, \mathbf{p}) S^{\Lambda}(\Omega - \nu, \mathbf{q} - \mathbf{p}) + (G \leftrightarrow S), \quad (20b)$$

where $f_n$, $f_{n'}$ are form factors. The propagator includes the self-energy via the Dyson equation

$$G^{\Lambda}(k) = \left[\left[G_0^{\Lambda}(k)\right]^{-1} - \Sigma^{\Lambda}(k)\right]^{-1} = \left[\left(\frac{\Theta^{\Lambda}(k)}{i\nu - \epsilon_{\mathbf{k}} + \mu}\right)^{-1} - \Sigma^{\Lambda}(k)\right]^{-1}, \quad (21)$$

while the single-scale propagator is defined as $S^{\Lambda} = \partial_{\Lambda} G^{\Lambda}|_{\Sigma=\mathrm{const}}$. We note that the derivative is taken only on the explicit $\Lambda$-dependence of the propagator introduced by the cutoff function $\Theta^{\Lambda}(k)$. For the flow of the self-energy we use the conventional $1\ell$ equation, see Ref. [20] for the details. We note that for $\lambda^{\mathrm{X}} = \mathbb{1}$, the flow equation for the bosonic propagator can be integrated, yielding the RPA. Neglecting the flow of the rest function $M^{\mathrm{X}}$ is referred to as the SBE approximation[3].

We now come to the differences in the generalised SBE for extended interactions. First, the initial conditions for the flowing objects are given by

$$w^{\mathrm{X,init}}(\Omega, \mathbf{q}) = \mathcal{B}^{\mathrm{X}}(\mathbf{q}) \tag{22a}$$

$$\lambda^{\mathrm{X,init}}(\Omega, \mathbf{q}, \nu, n) = \begin{cases} 1 & s\text{-wave} \\ 0 & \text{else} \end{cases} \tag{22b}$$

$$M^{\mathrm{X,init}}(\Omega, \mathbf{q}, \nu, n, \nu', n') = 0. \tag{22c}$$

---

[3]Neglecting the *flow* of $M^{\mathrm{X}}$ does not imply that all multiboson processes are neglected. In fact, contributions to multiboson processes will still be generated during the flow. In [54], this approximation is therefore termed to as the SBE$a$ approximation to distinguish it from the more restrictive SBE$b$ approximation where multiboson exchanges are completely neglected.

It follows that the $\mathcal{B}$-irreducible vertex in a specific channel fulfills

$$\mathcal{I}^{\mathrm{X,init}} = \mathcal{F}^{\mathrm{X}}. \tag{23}$$

The second difference consists in the flows of the non-trivial asymptotics objects for the fermion-boson coupling and the rest function, which has in principle to be considered

$$\lambda^{\mathrm{X,asympt}}(\Omega, \mathbf{q}, n) = \lim_{\nu \to \infty} \lambda^{\mathrm{X}}(\Omega, \mathbf{q}, \nu, n), \tag{24a}$$

$$M_2^{\mathrm{X}}(\Omega, \mathbf{q}, n, \nu', n') = \lim_{\nu \to \infty} M^{\mathrm{X}}(\Omega, \mathbf{q}, \nu, n, \nu', n'), \tag{24b}$$

$$M_1^{\mathrm{X}}(\Omega, \mathbf{q}, n, n') = \lim_{\nu \to \infty, \nu' \to \infty} M^{\mathrm{X}}(\Omega, \mathbf{q}, \nu, n, \nu', n'). \tag{24c}$$

The corresponding flow equations, Eqs. (19b) and (19c), determine the respective high-frequency limits. Moreover, since the additional momentum dependence stems from the extended interaction, it will involve only the bond form-factor indices corresponding to the range of the interaction. In other words, if the bare interaction has a finite range in space, then the non-trivial asymptotics is trivial outside.

In the application to the extended Hubbard model (1), we use a smooth frequency cutoff, supplementing the $\Omega$-flow with a flowing chemical potential for a computation at fixed filling

$$G_0^{\Lambda}(i\nu, \mathbf{k}) := \frac{\Theta^{\Lambda}(i\nu)}{G_0^{-1}(i\nu, \mathbf{k}) + \delta\mu^{\Lambda}}, \tag{25}$$

with $\Theta^{\Lambda}(i\nu) = \nu^2/(\nu^2 + \Lambda^2)$ and $\delta\mu_{\Lambda}$ a scale-dependent chemical potential shift. The single-scale propagator then reads

$$S = G^{\Lambda} \left[ \dot{\Theta}^{\Lambda}(G_0^{-1} + \delta\mu^{\Lambda}) - \Theta^{\Lambda}\delta\dot{\mu}^{\Lambda} \right] G^{\Lambda}. \tag{26}$$

The chemical potential is adjusted to counter the flow of the filling. For a specified filling $n$, the shift of the chemical potential is determined by inverting the function [15]

$$n(\delta\mu^{\Lambda}) = \sum_{\nu} \int d\mathbf{k} \frac{e^{i\nu0^+}}{G_0^{-1}(i\nu, \mathbf{k}) + \delta\mu^{\Lambda} - \Theta^{\Lambda}(i\nu)\Sigma^{\Lambda}(i\nu, \mathbf{k})}. \tag{27}$$

Numerically, this is done by a root finding algorithm on the equation $n - n(\delta\mu^{\Lambda}) = 0$. For the $j$-th flow integration step $\Lambda_j$, the derivative is calculated by the finite difference

$$\delta\dot{\mu}^{\Lambda_{j+1}} = \frac{\delta\mu^{\Lambda_{j+1}} - \delta\mu^{\Lambda_j}}{\Lambda_{j+1} - \Lambda_j}, \tag{28}$$

used for its feedback on the single-scale propagator.

In Table 1 we report the technical parameters. For testing the algorithmic approximations, we used $N_w = 5$ for the frequencies and $N_k = 16$, $N_{k,\delta} = 0$ for the momenta, where $N_{k,\delta}$ specifies the refinement around $(\pi, \pi)$ as illustrated in Fig. 4. For the calculation of the physical susceptibilities and their analysis, we used $N_w = 6$, $N_k = 18$ and a finite $N_{k,\delta}$. The increase in the overall resolution in the latter is made viable by the approximations discussed in Section 4.1.

|       | Bosonic frequencies | Fermionic frequencies | Bosonic momenta | Fermionic momenta |
|-------|---------------------|-----------------------|-----------------|-------------------|
| $w$     | $128N_w + 1$        | $\varnothing$         | $N_k^2 + N_{k,\delta}$ | $\varnothing$     |
| $\lambda$ | $4N_w + 1$        | $2N_w$                | $N_k^2 + N_{k,\delta}$ | form factors      |
| $M$     | $4N_w + 1$          | $2N_w$                | $N_k^2 + N_{k,\delta}$ | form factors      |
| $\Sigma$ | $\varnothing$      | $8N_w$                | $\varnothing$   | $N_k^2$           |
| $\Pi$   | $128N_w + 1$        | $128N_w$              | $N_k^2 N_\delta^2$ | form factors   |

Table 1: Number of frequency and momentum components used for the parametrization, where $N_\delta = 5$ for the bubble. For the treatment of the momenta within the TU-fRG, the bosonic momenta of the vertices live on a discretized Brillouin zone, whereas the fermionic momentum dependence is restricted to (a few) form factors. Specifically, we use only an $s$-wave form factor at half filling, and include an additional $d$-wave one at finite doping. The fermionic momenta of the self-energy live on a discretized Brillouin zone.

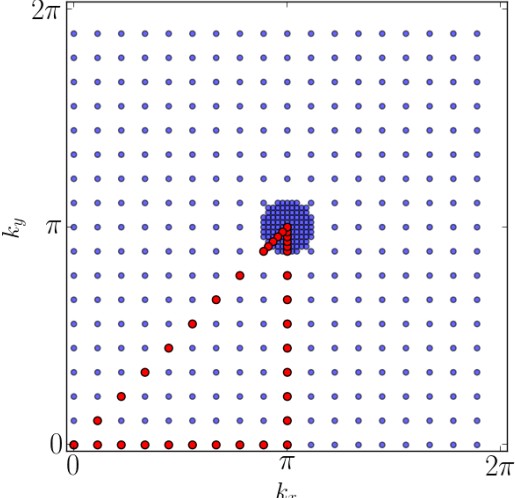

Figure 4: The momentum grid with the refinement around $(\pi, \pi)$ used for the bosonic momentum dependence of $w$, $\lambda$, and $M$. The additional $N_{k,\delta}$ points allow to resolve the antiferromagnetic and charge-density wave peaks that appear at and in the proximity of $(\pi, \pi)$. The points in red mark the path $\Gamma$-$X$-$M$-$\Gamma$.

### 3.4 Fluctuation diagnostics

The fluctuation diagnostics [79, 80] allows to identify the fluctuations at play and responsible for the physical behavior. By inserting the parquet decomposition of the vertex, its contributions provide information on the effects of different fluctuation channels to physical observables, see Refs. [57, 71, 81, 82] for applications to the self-energy and susceptibilities.

The susceptibility can be expressed as a sum of two terms

$$\chi^{X} = \chi^{X}_{\text{bubble}} + \chi^{X}_{\text{vertex}}, \tag{29}$$

where the first denotes the bubble and represents the bare susceptibility with self-energy corrections

$$\chi^{X}_{\text{bubble}}(\Omega, \mathbf{q}, n, n') = \sum_{\nu} \Pi^{X}(\Omega, \mathbf{q}, \nu, n, n') \tag{30}$$

and the second term the vertex contributions

$$\chi^{X}_{\text{vertex}}(\Omega, \mathbf{q}, n, n') = -\sum_{m,m',\nu,\nu'} \Pi^{X}(\Omega, \mathbf{q}, \nu, n, m) V^{X}(\Omega, \mathbf{q}, \nu, m, \nu', m') \Pi^{X}(\Omega, \mathbf{q}, \nu', m', n'). \tag{31}$$

Here the vertex and self-energy used are the ones obtained at the end of the fRG flow, see also Refs. [13, 83] for an application. This idea can be extended to the SBE decomposition of the vertex [13, 84]. In order to avoid double-counting contributions which have no physical meaning, we introduce

$$\bar{\nabla}^{X} \equiv \nabla^{X} - U^{X} \tag{32}$$

subtracting the bare local interaction (see also Appendix A). This determines the decomposition

$$V^{X} = V^{X}_{\bar{\nabla}M} + V^{X}_{\bar{\nabla}D} + V^{X}_{\bar{\nabla}SC} + V^{X}_{V_{\text{irr}}} + U^{X}, \tag{33}$$

where $V^{X}_{\bar{\nabla}X'}$ includes all SBE contributions to $V^{X}$, i.e. that involve $\bar{\nabla}^{X'}$. The interaction irreducible contribution $V_{V_{\text{irr}}}$ contains all contributions of $\bar{M}^{X}$ and $\bar{V}_{2\text{PI}}$, except for $U^{X}$ ($V_{V_{\text{irr}}} = 0$ in the parquet approximation). Inserting this splitting into Eq. (31), we obtain

$$\chi^{X} = \chi^{X}_{\text{bubble}} + \chi^{X}_{\bar{\nabla}M} + \chi^{X}_{\bar{\nabla}D} + \chi^{X}_{\bar{\nabla}SC} + \chi^{X}_{V_{\text{irr}}} + \chi^{X}_{U}, \tag{34}$$

where $\chi^{X}_{\bar{\nabla}X'}$ denotes the SBE contribution from $V^{X}_{\bar{\nabla}X'}$, $\chi^{X}_{V_{\text{irr}}}$ the multiboson contributions from $V_{V_{\text{irr}}}$, and $\chi^{X}_{U}$ the contribution from the bare local interaction $U$.

In the SBE formulation, it is useful to introduce the polarization obtained from the fermion-boson vertex by

$$P^{X}(\Omega, \mathbf{q}) = \sum_{m,\nu} \lambda^{X}(\Omega, \mathbf{q}, \nu, m) \Pi^{X}(\Omega, \mathbf{q}, \nu, m, n = s\text{-wave}). \tag{35}$$

It is related to the bosonic propagator $w^{X}$, which in this context we refer to as the screened interaction, by

$$w^{X} = \frac{\mathcal{B}^{X}}{1 + \mathcal{B}^{X} P^{X}}. \tag{36}$$

From Eq. (35), it is evident that a deviation from $\lambda(m = s\text{-wave}) = 1$ implies that the screened interaction departs from the RPA prediction.

Analogously, the fluctuation diagnostics can be performed on the fermion-boson coupling and the polarization. From the definition [53, 85–87]

$$\lambda^{\mathrm{X}}(\Omega, \mathbf{q}, \nu, n) = \delta(n, s\text{-wave}) - \sum_{\nu', n'} \mathcal{I}^{\mathrm{X}}(\Omega, \mathbf{q}, \nu, n, \nu', n') \Pi^{\mathrm{X}}(\Omega, \mathbf{q}, \nu', n', n), \qquad (37)$$

we infer that the fluctuation diagnostics relies on the decomposition of $\mathcal{I}^{\mathrm{X}}$. Since $\mathcal{I}^{\mathrm{X}} = V^{\mathrm{X}} - \nabla^{\mathrm{X}} = V^{\mathrm{X}} - \bar{\nabla}^{\mathrm{X}} - U^{\mathrm{X}}$, we find

$$\mathcal{I}^{\mathrm{X}} = \mathcal{I}_{\bar{\nabla}}^{\mathrm{X}} + \sum_{\mathrm{X}' \neq \mathrm{X}} \mathcal{I}_{\bar{\nabla}\mathrm{X}'}^{\mathrm{X}} + \mathcal{I}_{V_{\mathrm{irr}}}^{\mathrm{X}}, \qquad (38)$$

where $\mathcal{I}_{V_{\mathrm{irr}}}^{\mathrm{X}} = V_{V_{\mathrm{irr}}}^{\mathrm{X}}$, $\mathcal{I}_{\bar{\nabla}\mathrm{X}'}^{\mathrm{X}} = V_{\bar{\nabla}\mathrm{X}'}^{\mathrm{X}}$, and $\mathcal{I}_{\bar{\nabla}}^{\mathrm{X}} = V_{\bar{\nabla}}^{\mathrm{X}} - \bar{\nabla}^{\mathrm{X}}$. Note that there is no contribution from the bare local interaction $U^{\mathrm{X}}$. Inserting it into Eq. (37), we get

$$\lambda^{\mathrm{X}} = \delta(n, s\text{-wave}) + \lambda_{\bar{\nabla}}^{\mathrm{X}} + \sum_{\mathrm{X}' \neq \mathrm{X}} \lambda_{\bar{\nabla}\mathrm{X}'}^{\mathrm{X}} + \lambda_{\mathrm{Virr}}^{\mathrm{X}}, \qquad (39)$$

which in Eq. (35) yields

$$P^{\mathrm{X}} = P_{\mathrm{bubble}}^{\mathrm{X}} + P_{\bar{\nabla}}^{\mathrm{X}} + \sum_{\mathrm{X}' \neq \mathrm{X}} P_{\bar{\nabla}\mathrm{X}'}^{\mathrm{X}} + P_{\mathrm{Virr}}^{\mathrm{X}}. \qquad (40)$$

Some contributions to the polarization are directly related to the ones of the $s$-wave susceptibility

$$P_{\mathrm{bubble}}^{\mathrm{X}} = \chi_{\mathrm{bubble}}^{\mathrm{X}}(m = s\text{-wave}, n = s\text{-wave}) \qquad (41a)$$

$$P_{\mathrm{Virr}}^{\mathrm{X}} = \chi_{\mathrm{Virr}}^{\mathrm{X}}(m = s\text{-wave}, n = s\text{-wave}) \qquad (41b)$$

and for $X \neq X'$

$$P_{\bar{\nabla}\mathrm{X}'}^{\mathrm{X}} = \chi_{\bar{\nabla}\mathrm{X}'}^{\mathrm{X}}(m = s\text{-wave}, n = s\text{-wave}). \qquad (42)$$

Finally, we mention that this "post-processing" analysis will be performed with the fRG results at the end of the flow and applies during the flow only within a two-particle self-consistent truncation such as the multiloop scheme. This leads to small discrepancies between the sums of the contributions obtained from the decompositions above and the results obtained from the $1\ell$ fRG flow of the corresponding response functions [5]. We have verified, however, that multiloop corrections do not alter our conclusions.

## 4   Results

We first provide a systematic study of the various approximations together with their impact on the algorithmic implementation and then compute the physical susceptibilites of the extended Hubbard model, both at half filling as well as at finite doping. A detailed fluctuation diagnostics of their evolution with $U'$ allows us to determine the driving fluctuation channels.

### 4.1   A handy computation scheme

The presence of a nonlocal interaction leads to a more involved numerical treatment, especially the larger number of form factors together with the additional non-trivial high-frequency asymptotics, which results in a substantial increase of the run times. Approximations are necessary to venture into the physically interesting parameter regimes. We

here inspect a series of approximations, assessing their impact on the accuracy. For simplicity we consider $t' = 0$ at half filling ($n = 0.5$), $U = 2$, $U' = U/4$, and $\beta = 5$ for this analysis, unless otherwise stated. It corresponds to a situation on the verge to the $U'$-induced transition from dominant antiferromagnetic fluctuations to a charge-density wave instability at low temperatures, where $s$-wave charge fluctuations dominate (see Fig. 5). This is expected to occur around $U' = U/4$.

We first observe that the so-called mixed bubbles $\Pi_{nn'}$ with $n \neq n'$ are in general significantly smaller than the ones diagonal in form factors, i.e. with $n = n'$. This allows us to restrict the summation over form factors on the right-hand side of the flow equations to

$$\dot{w}^{\mathrm{X}}(\Omega, \mathbf{q}) = -\mathrm{Sgn}\, \mathrm{X}\, w^{\mathrm{X}}(\Omega, \mathbf{q}) \sum_{\nu n n'} \lambda^{\mathrm{X}}(\Omega, \mathbf{q}, \nu, n) \dot{\Pi}^{\mathrm{X}}(\Omega, \mathbf{q}, \nu, n, n') \lambda^{\mathrm{X}}(\Omega, \mathbf{q}, \nu, n') w^{\mathrm{X}}(\Omega, \mathbf{q}),$$

(43a)

$$\approx -\mathrm{Sgn}\, \mathrm{X}\, w^{\mathrm{X}}(\Omega, \mathbf{q}) \sum_{\nu n} \lambda^{\mathrm{X}}(\Omega, \mathbf{q}, \nu, n) \dot{\Pi}^{\mathrm{X}}(\Omega, \mathbf{q}, \nu, n, n) \lambda^{\mathrm{X}}(\Omega, \mathbf{q}, \nu, n) w^{\mathrm{X}}(\Omega, \mathbf{q}),$$

(43b)

which scales only linearly in the number of form factors. Figure 5 illustrates the minimal effect this approximation has on the results for the bosonic propagator, with deviations below 0.1%. The fermion-boson couplings are even less affected, since this approximation

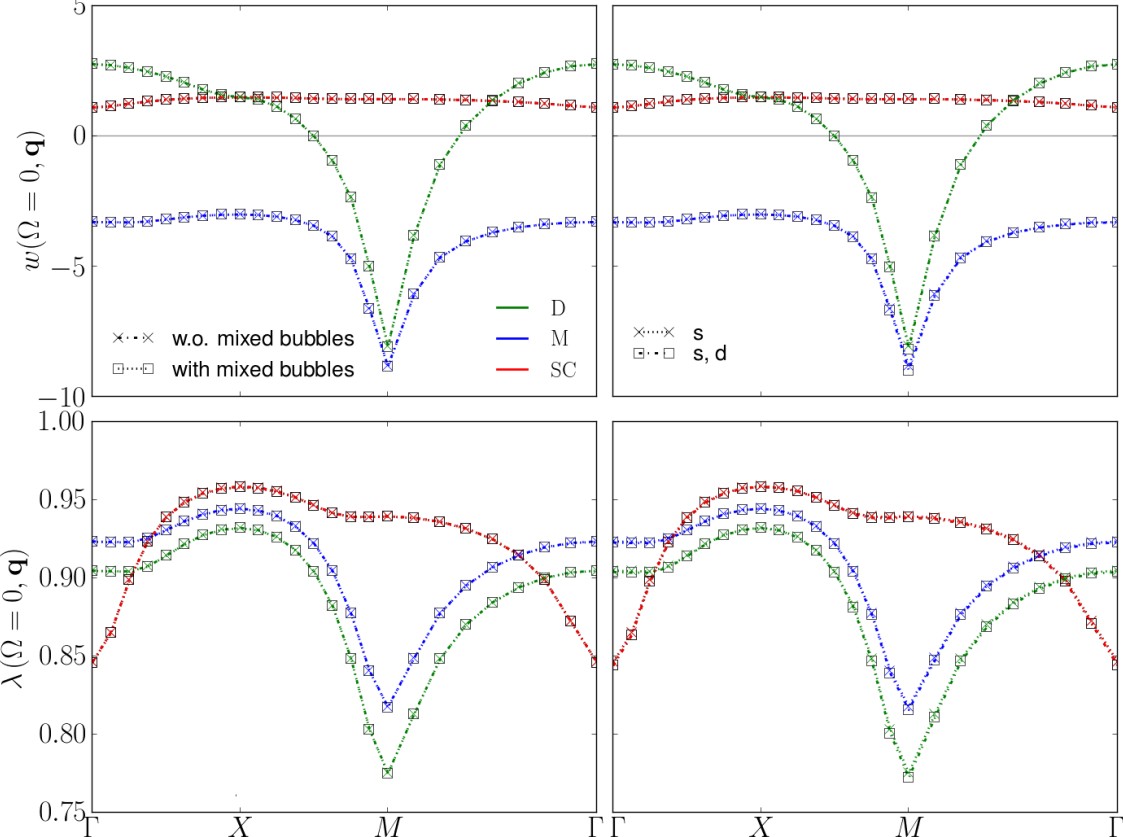

Figure 5: Bosonic propagator $w^{\mathrm{X}}$ and ($s$-wave) fermion-boson coupling $\lambda^{\mathrm{X}}$ as obtained from the calculation with and without mixed bubbles (left column) as well as with increasing number of form factors (right column), for $U = 2$, $U' = U/4$, $T = 0.2$, at half filling.

enters only indirectly. Throughout the present study, we will hence use this approximation for all further calculations (the run times for the current implementation including mixed bubbles are numerically also not feasible). We thereby assume that the approximations discussed in the following do not alter its validity notably.

In a straightforward fRG application to the model as described in Appendix B, the increased numerical effort with respect to the Hubbard model with only a local interaction is due to the larger number of form factors that have to be considered. There, in order to correctly account for a nearest-neighbor interaction $U'$, the full first shell, i.e. five form factors (see Ref. [20] for their definition) is required to expand $V_0$ in the $\overline{ph}$- and $pp$-channel without information loss. In contrast, in the proposed SBE extension, where the interaction is decomposed into a bosonic and a fermionic part, only the fermionic part $\mathcal{F}^{\mathrm{X}}$ requires a form-factor expansion. In this case, the effect of a form-factor truncation turns out to be very small and a quantitatively accurate description is obtained by taking into account only the local $s$-wave one, see Fig. 5. Results including further form factors beyond $d$-wave can be found in Ref. [20], while the effect of the non-trivial high-frequency asymptotics that can arise is discussed in Appendix D.

In the following, we provide an analytical understanding of these findings. In the $ph$-channel the bare extended Hubbard interaction is captured by an $s$-wave form factor, whereas the $\overline{ph}$- and $pp$-channel have no $s$-wave contribution. The flow of the bosonic propagator described by Eq. (43a) favors $s$-wave contributions in two ways: i) mixed bubbles $\Pi_{nn'}$ with $n \neq n'$ are small and ii) the $s$-wave components of $\lambda^{\mathrm{X}}$ are of $\mathcal{O}(1)$, whereas all others are initialized to zero and remain small throughout the flow. Therefore, the nonlocal contributions to $\dot{w}^{\mathrm{X}}$ are either suppressed by mixed bubbles or by vanishing nonlocal $\lambda^{\mathrm{X}}$. A similar picture emerges for the $s$-wave fermion-boson couplings, which appear to be accurately described by a calculation neglecting all nonlocal form factors. From the structure of the equation

$$\dot{\lambda}^{\mathrm{X}}(\Omega,\mathbf{q},\nu,s\text{-wave}) = -\operatorname{Sgn}\mathrm{X} \sum_{\nu' n n'} \mathcal{I}^{\mathrm{X}}(\Omega,\mathbf{q},\nu,s\text{-wave},\nu',n)\dot{\Pi}^{\mathrm{X}}(\Omega,\mathbf{q},\nu',n,n')\lambda^{\mathrm{X}}(\Omega,\mathbf{q},\nu,n')$$

(44)

one sees that the same arguments as above apply also here. We note that the restriction to a local form factor implies also that the non-trivial asymptotics can be omitted, since these occur only for nonlocal form factors.

The physically intuitive picture of the effects of a nearest-neighbor interaction can be underpinned by analytical insights: the nonlocal interaction directly affects the charge channel and hence the $s$-wave charge susceptibility, which is expected to exhibit the strongest dependence on $U'$. All other susceptibilities are altered only indirectly. This is reflected by the nearest-neighbor interaction appearing purely bosonic in particle-hole notation, where it directly enters the equations for the bosonic propagator and the $s$-wave fermion-boson coupling of the density channel, in contrast to the ones for the $pp$ and $\overline{ph}$ channels where its influence is structurally suppressed. A natural interpretation of this finding is that the bosonic propagator encodes the main ordering tendencies. It highlights how the SBE decomposition serves not only to reduce the numerical effort but also to gain a deeper understanding of the fluctuations controlling the physical behavior.

We now analyse the validity of the SBE approximation, i.e. neglecting the flow of the rest functions, in presence of a finite $U' > 0$. The results for the static susceptibilities and the fermion-boson couplings obtained with and without rest functions are shown in Fig. 6. Almost down to the lowest temperature $T = 0.1$, the SBE approximation yields quantitatively accurate results. This holds also at finite frequencies (not shown). We remark that although the influence of the rest function on the $s$-wave susceptibilities remains small, its absolute values may not be small.

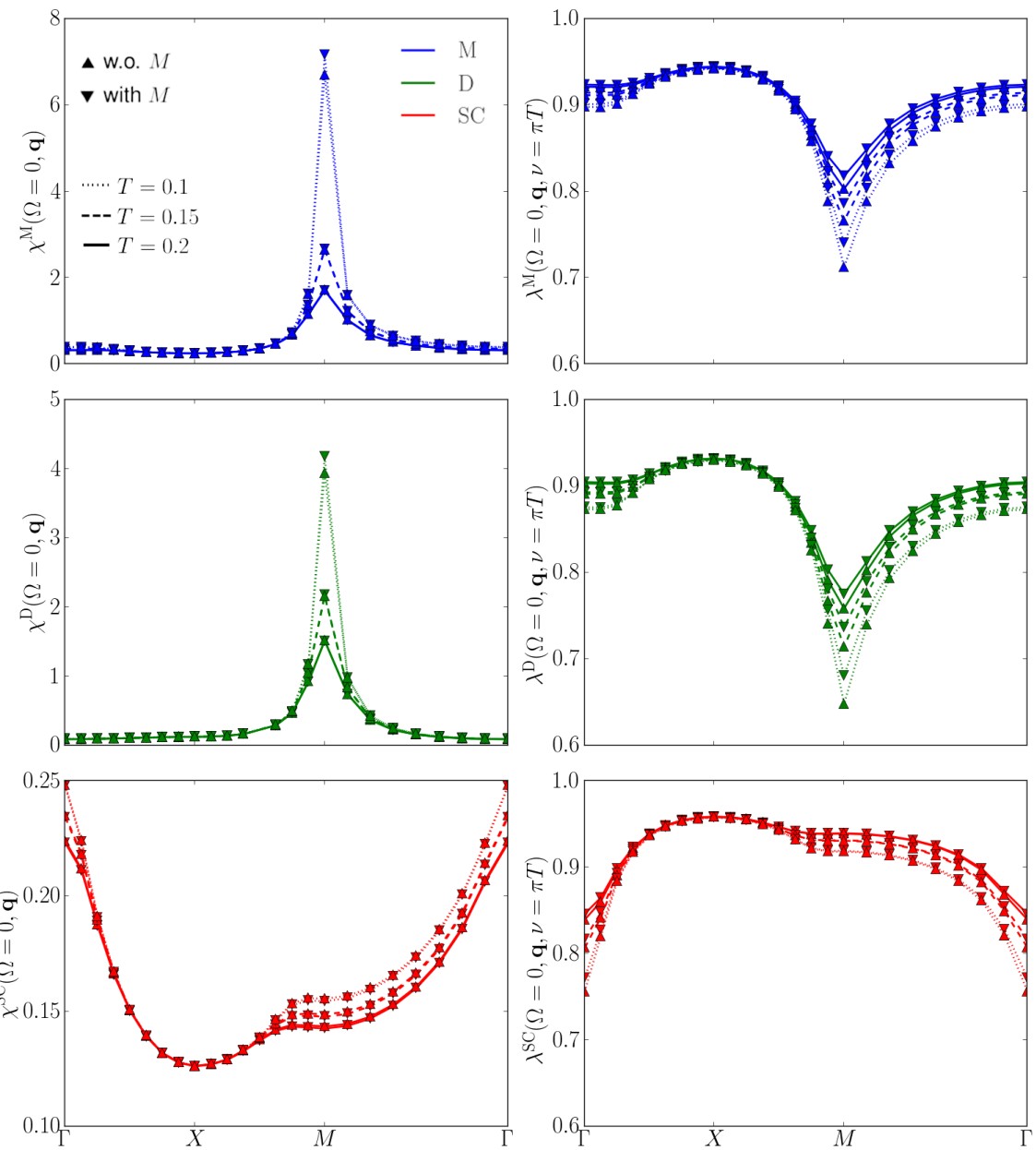

Figure 6: Momentum dependence of $\chi^{\text{X}}$ and $\lambda^{\text{X}}$ as obtained from a calculation with and without the rest function, for the same parameters as in Fig. 5 and various temperatures (using only an $s$-wave form factor). The relative difference between the fermion-boson coupling $\lambda^{\text{X}}$ calculated with and without the flow of the rest function is below 2 % in all channels including the lowest temperature $T = 0.1$. The same holds for $\chi^{\text{SC}}$ at all temperatures and $\chi^{\text{D/M}}$ above $T = 0.1$. At $T = 0.1$ both the magnetic and charge channel are approaching a divergence and the relative difference increases to 6 % and 8 % respectively, comparable to the behavior in the Hubbard model [54].

Also at finite doping, these approximations turn out to provide an accurate description. We focus on $t' = -0.2$ at van Hove filling $\mu = 4t'$. Here, the flow of the chemical potential is of major relevance and has to be accounted for in fixing the filling. As previously, we use the value at $U = U' = 0$, corresponding to $n = 0.415$. We note that the increase of the computational cost is significant and is reflected in the increased run times with respect to calculations where the filling is renormalized by the flow. In Fig. 7, we show the results for the same parameters considered previously, but at finite doping. The effect of an additional $d$-wave form factor as well as of the contributions of bubbles mixing $s$- and $d$-wave components of the vertices are negligible also in this case, the data perfectly match the only $s$-wave ones. This is surprising, since at finite doping $d$-wave fluctuations are expected to play an important role in the Hubbard model. However, these typically arise at lower temperatures. We here aim at studying the interplay between antiferromagnetic and charge-density wave fluctuations and therefore refrain from their analysis. Figure 8 displays the results obtained from the computation with and without rest function flow at various temperatures. The overall picture is similar: while neglecting the rest function leads to small deviations in the fermion-boson coupling, the susceptibilities appear hardly affected. The differences are in fact of the same order as the ones at half filling. We therefore conclude that the proposed scheme can be applied also at finite doping.

Summarizing, we established a simplified computation scheme with only an $s$-wave form factor, that neglects the rest functions as well as the high-frequency asymptotics

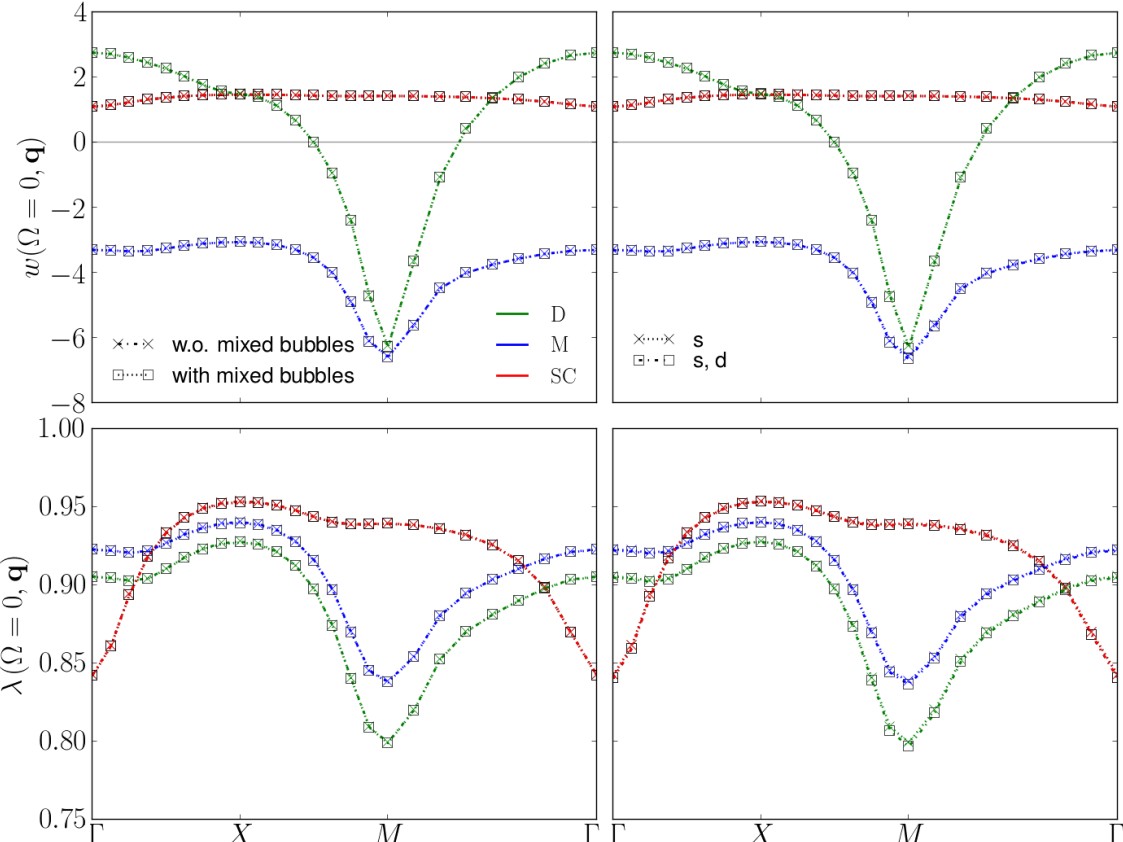

Figure 7: Bosonic propagator $w^{\mathrm{X}}$ and ($s$-wave) fermion-boson coupling $\lambda^{\mathrm{X}}$ as obtained from the calculation with and without mixed bubbles (left column) as well as with increasing number of form factors (right column), for $U = 2$, $U' = U/4$, $T = 0.2$, at van Hove filling with $t' = 0.2t$.

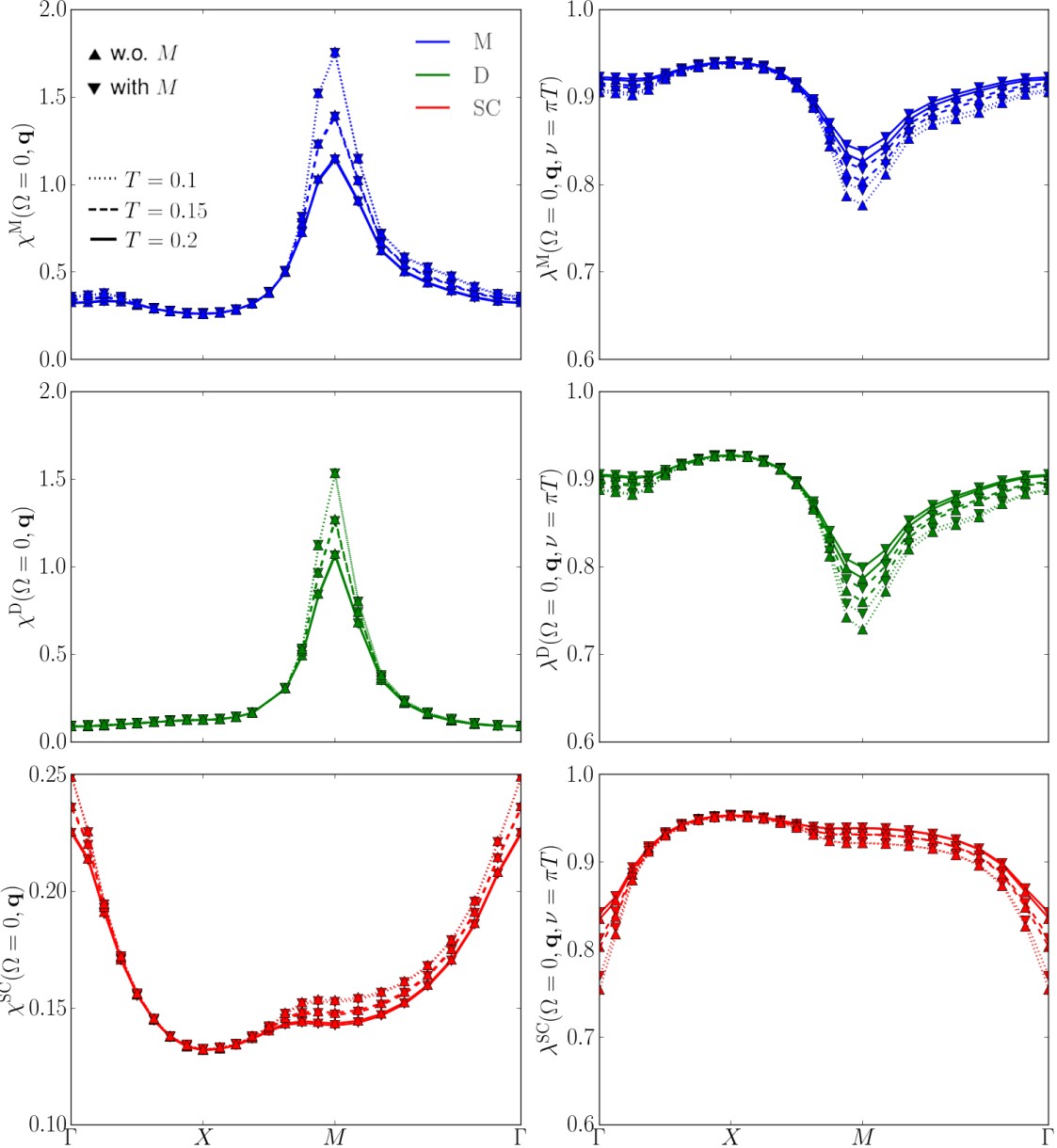

Figure 8: $\chi^{\mathrm{X}}$ and $\lambda^{\mathrm{X}}$ as obtained from a calculation with and without the rest function, for the same parameters as in Fig. 6 but at van Hove filling corresponding to $t' = -0.2$ (using only an $s$-wave form factor).

of the fermion-boson couplings. The reduced run times, comparable to the ones for the Hubbard model ($U' = 0$), allow for a systematic investigation of the effects induced by an additional nearest-neighbor interaction. We first focus on the physical behavior at half filling, where we determine the physical susceptibilities in the different channels and analyse the results with a fluctuation diagnostics.

## 4.2 Analysis of the extended Hubbard Model

We now investigate the influence of $U'$ on the various SBE quantities, starting with the analysis of the half-filled model. Figure 9 displays the evolution of $w^X$ and $\lambda^X$ with $U'$, for $U = 2$ and $\beta = 10$ as representative parameters for the weak-coupling regime. With increasing $U'$, the fermion-boson coupling $\lambda^M$ decreases, while $\lambda^{SC}$ increases. In contrast, the density coupling $\lambda^D$ appears almost unaffected. We emphasize that for a clear observation on this effect, the filling has to be kept constant (it flows in presence of a finite $U'$ since particle-hole symmetry is broken). For the susceptibilities, Fig. 9 illustrates the competition between the antiferromagnetism and charge-density wave order: while $\chi^M$ is reduced with $U'$, $\chi^D$ is enhanced, until it diverges around $U' = U/4$. It is remarkable

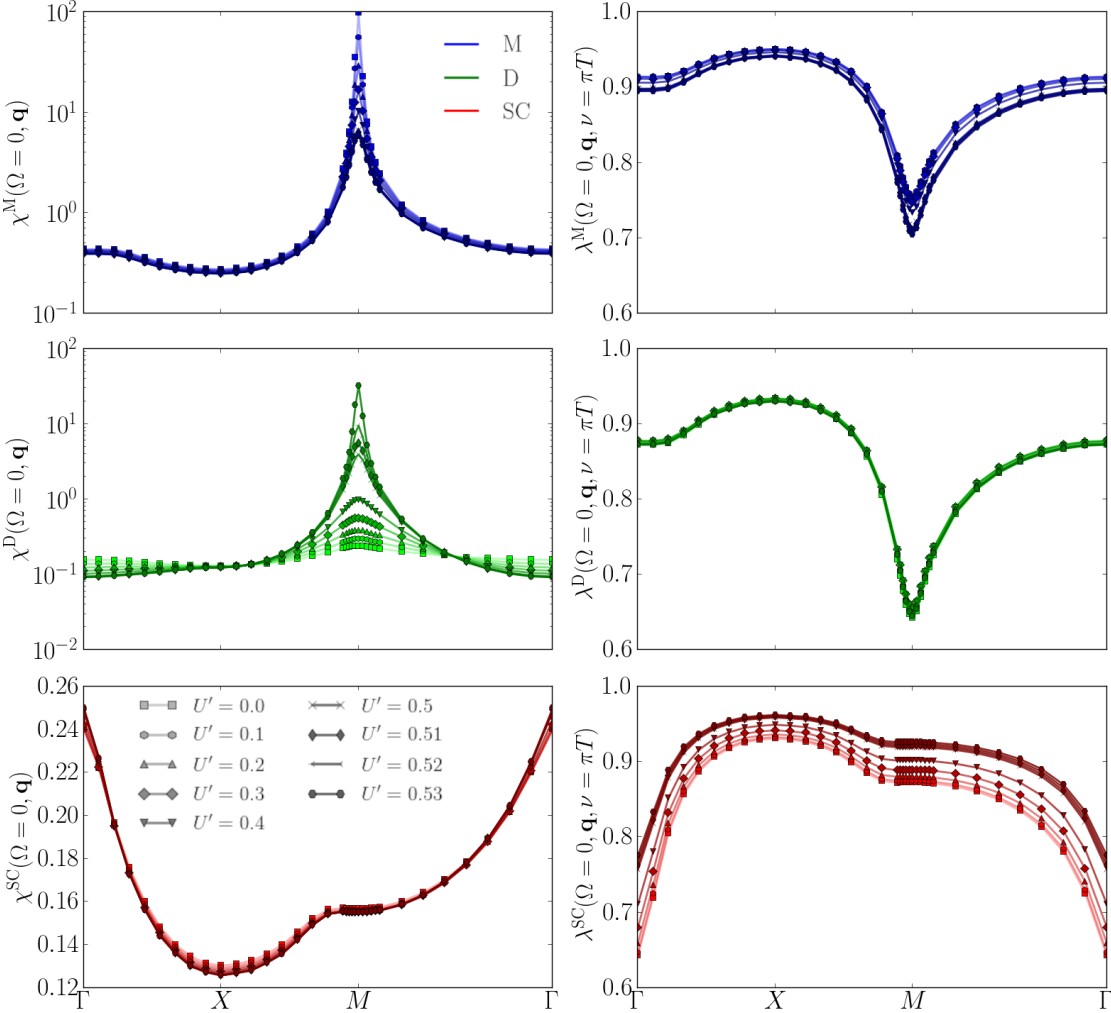

Figure 9: Momentum dependence of $\chi^X$ and $\lambda^X$ and their evolution with $U'$, for $U = 2$, $\beta = 10$, and at half filling ($t' = 0$). Note the logarithmic scale in the susceptibility panels for magnetic and density channels.

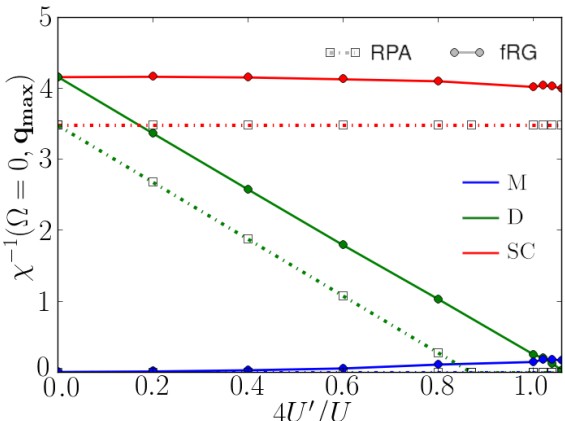

Figure 10: Maximal values of the magnetic, charge (or density) and superconducting susceptibilities as a function of $U'$, for $U = 2$, $\beta = 10$, and at half filling ($t' = 0$), together with the RPA susceptibilities for comparison.

that despite the drastic increase of $\chi^{\mathrm{D}}$, $\lambda^{\mathrm{D}}$ is almost independent of $U'$. The ($s$-wave) superconducting susceptibility varies only slightly with $U'$.

The maximal values of the static susceptibilities $\chi^{\mathrm{M}}(\Omega = 0, \mathbf{q} = (\pi, \pi))$, $\chi^{\mathrm{D}}(\Omega = 0, \mathbf{q} = (\pi, \pi))$, and $\chi^{\mathrm{SC}}(\Omega = 0, \mathbf{q} = 0)$ and their evolution with $U'$ are shown in Fig. 10. For comparison, we also report the results of the RPA. At $U' = 0$, $\chi^{\mathrm{M}}$ is close to the antiferromagnetic divergence, all other susceptibilities are orders of magnitude smaller. For a finite $U'$, $\chi^{\mathrm{M}}$ decreases whereas $\chi^{\mathrm{D}}$ increases with $U'$. They cross around $4U'/U = 1$, before $\chi^{\mathrm{D}}$ diverges at $4U'/U = 1.1$. $\chi^{\mathrm{SC}}$ is an order of magnitude smaller than $\chi^{\mathrm{M}}$ and only slightly increases with $U'$. For a finite $U'$, the particle-hole symmetry between $\chi^{\mathrm{D}}(\mathbf{q} = (\pi, \pi))$ and $\chi^{\mathrm{SC}}(\mathbf{q} = (0, 0))$ at half filling [16] is lifted. Physically, the crossover from the dominant antiferromagnetic peak to a charge-density wave instability observed for $4U' \sim U$ can be understood by a simple picture of the strong-coupling regime: at half filling, the energy cost per site of a perfectly even charge distribution is $l \cdot U'$ in the atomic limit, where $l$ is the number of bonds per site ($l = 4$ for the square lattice). At the same time, the average cost per site for a charge-density wave with $\mathbf{q} = (\pi, \pi)$ is given by $U$, see Fig. 11. The compensation of these two competing effects determines the crossover. Upon further increasing $U'$, the charge susceptibility diverges, indicating the onset of a strong tendency to a charge-density wave order. The RPA data exhibit a qualitatively similar behavior for the charge and superconducting susceptibilities. Quantitatively, the absolute

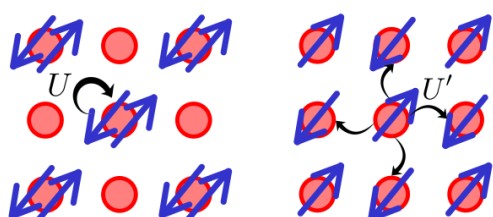

Figure 11: Competition between charge-density wave (left) and antiferromagnetic (right) configurations at half filling. The interaction energy cost per electron is $U$ in the charge-density wave configuration compared to $4U'$ per electron in the antiferromagnetic configuration.

values are enhanced due to the absence of screening. In particular, the inter-channel feedback encoded in the fermion-boson couplings set to 1 is not considered in RPA. The magnetic susceptibility, in contrast, diverges for all values of $U'$, since the interplay with the other channels is neglected. Specifically, the influence of the charge-density wave is missing.

We now consider a finite doping, with $n = 0.415$ corresponding to van Hove filling for $\mu = -0.8$ and $t' = -0.2$, for $U = 2$ and $\beta = 10$. The resulting susceptibilities and fermion-boson couplings for the different channels are reported in Fig. 12. Since the $d$-wave physics will become relevant only at lower temperatures, we here focus on the $s$-wave components (although the computations here include also the $d$-wave form factor). As expected, a finite $t'$ leads to a suppression of antiferromagnetic fluctuations, which is clearly seen in the reduced peak of $\chi^{\mathrm{M}}$. Note that in this case, the wave vector in correspondence of the maximum is incommensurate. The observed behavior of $\chi^{\mathrm{D}}$ and $\chi^{\mathrm{SC}}$ is qualitatively the same as for the half-filled case. In particular, we find that also at finite doping $\lambda^{\mathrm{D}}$ is almost unaffected by the nearest-neighbor interaction despite the significant enhancement induced in the corresponding susceptibility. This behavior will be further investigated

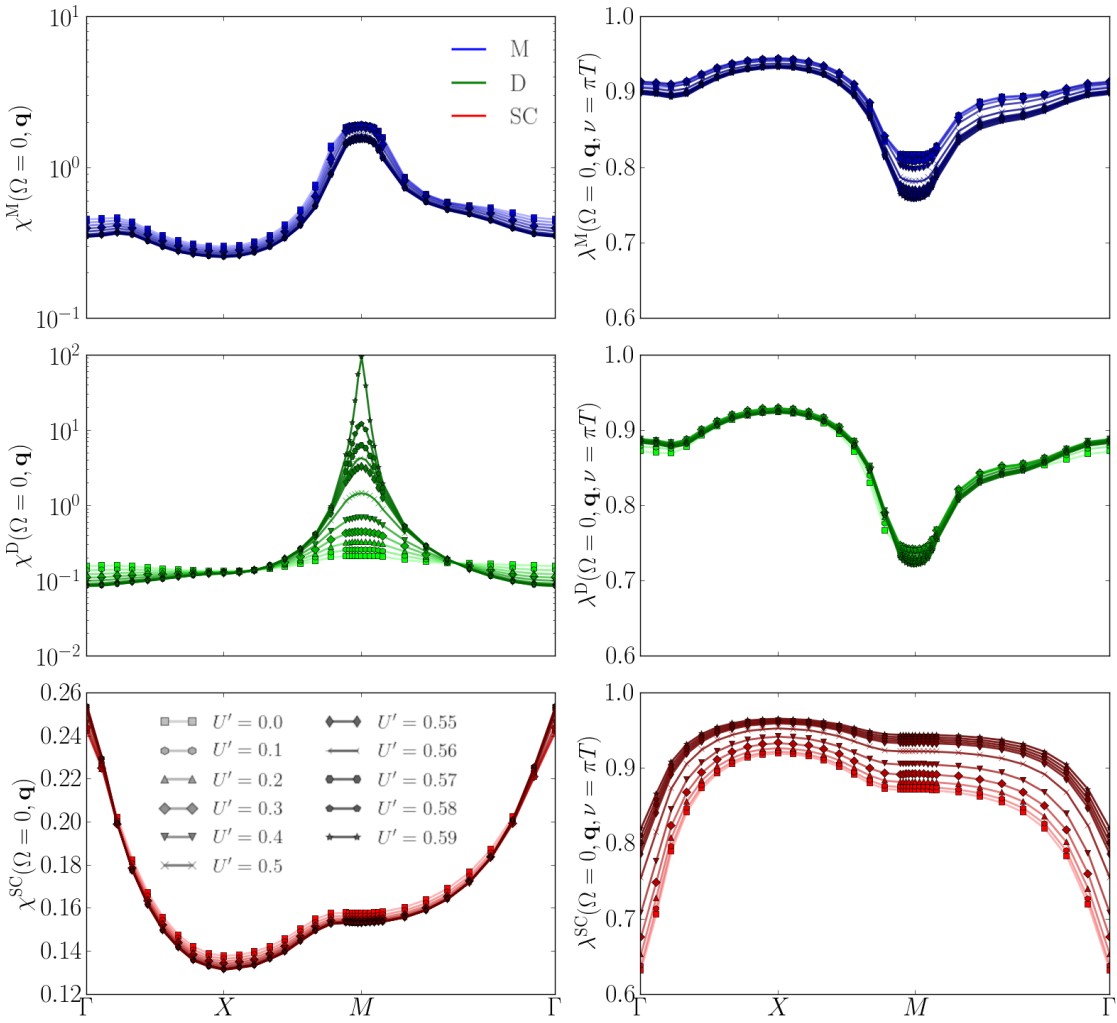

Figure 12: Momentum dependence of $\lambda^{\mathrm{X}}$ and $\chi^{\mathrm{X}}$ and their evolution with $U'$, for $U = 2$, $\beta = 10$, $t' = -0.2$, and at van Hove filling. Note the logarithmic scale in the susceptibility panels for magnetic and density channels.

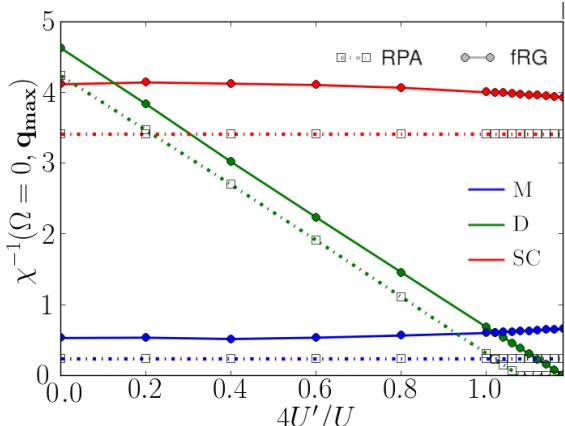

Figure 13: Maximal values of the magnetic, charge (or density) and superconducting susceptibilities, as a function of $U'$, for $U = 2$, $\beta = 10$, $t' = -0.2$, and at van Hove filling, together with the RPA susceptibilities for comparison.

within the fluctuation diagnostics in the next section. In analogy to Fig. 10, Fig. 13 displays the evolution of the maximal values of the susceptibilities with $U'$, now at finite doping. Also here, the most pronounced dependence is detected in $\chi^D$, its increase with $U'$ appears linear. In absence of perfect nesting, the magnetic RPA susceptibility acquires finite values. Otherwise, the overall trend is the same, with the RPA overestimating the susceptibilities.

For the RPA susceptibilities, the dependence on $U'$ can be understood by considering the relation $(\chi^X)^{-1} = (P^X)^{-1} + \mathcal{B}^X$, where $P^X$ is the polarization (35). In the RPA, the polarization is $P^X = \Pi_0^X$ and does not depend on the interaction. Thus, the linear decrease in the density channel is simply the one coming from $\mathcal{B}^D(\pi, \pi) = -8U'$, see Eq. (15). The almost flat curves for the superconducting and magnetic susceptibilities are thus due to the fact that the bosonic bare interactions $\mathcal{B}^{SC}$ and $\mathcal{B}^M$ are independent of $U'$. The fRG includes the interplay between the different channels, leading to a deviation from this behavior in the superconducting and magnetic susceptibilities. Remarkably, the charge susceptibility retains its linear dependence. This can be traced back to the insensitivity of $\lambda^D$ observed in Figs. 9 and 12. According to Eq. (35), this independence on $U'$ translates to the polarization $P^D$ [4]. This finding implies that the charge susceptibility of the extended Hubbard model can be approximated by

$$\chi^D \approx \frac{P_{U'=0}^D}{1 + P_{U'=0}^D \mathcal{B}^D}, \tag{45}$$

i.e. it can be described by an RPA-like formula with respect to the polarization of the Hubbard model at $U' = 0$. Importantly, the latter *does* include vertex corrections, leading to a quantitative deviation from the RPA. That is, negligible is only the dependence of $\lambda^D$ on $U'$, not $\lambda^D$ itself.

### 4.3   Fluctuation diagnostics

We now turn to the fluctuation diagnostics of the results presented above. This will provide a deeper insight on the effect of an extended interaction $U'$ on the interplay between the

---

[4]Note that inter-channel feedback in the fRG is encoded not only through $\lambda^X$, but also through self-energy corrections which are absent in RPA.

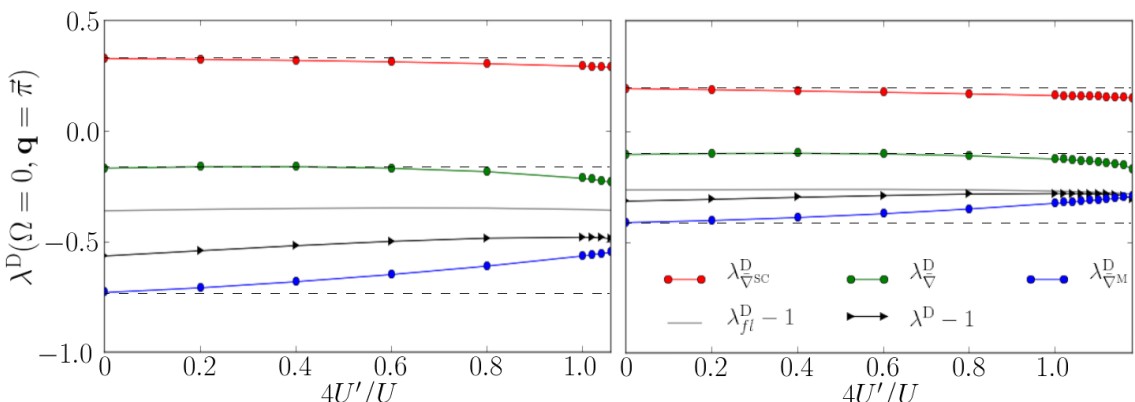

Figure 14: Fluctuation diagnostics of $\lambda^{\mathrm{D}}$ at half filling (left) and at finite doping (right), for the parameters of Figs. 9 and 12 (the dashed horizontal lines are guides to the eye). The contributions from the superconducting (red) and density (green) channels approximately cancel the contributions from the magnetic (red) one.

channels. According to the SBE decomposition defined in Section 3.4, we will consider post-processed results here.

In the following, we investigate the origin of the RPA-like behavior for the charge susceptibility. For this, we determine the different contributions to the density fermion-boson vertex $\lambda^{\mathrm{D}}$ at $\mathbf{q} = (\pi, \pi)$. In Fig. 14, we plot the results as a function of $U'$, both at half filling and at finite doping, for the parameters used previously in Figs. 9 and 12, respectively. The contributions from the $\mathcal{B}$-irreducible part which constitutes the rest function as well as the one from the two-particle irreducible part of the vertex are negligible and not shown. From the other contributions to $\lambda^{\mathrm{D}}$ we can identify the origin of the weak dependence on $U'$ responsible for the RPA-like behavior: it is caused by a cancellation between the contributions from the magnetic channel and the ones from (largely) the density and (to a smaller extent) superconducting channels. We note that the sum of the different contributions (black) obtained from Eq. (39) differs quantitatively from the flowing result (grey) retrieved from the bosonic propagator at the end of the fRG flow. In particular, the post-processing $\lambda^{\mathrm{D}}$ presents a slight variation with $U'$ as compared to $\lambda^{\mathrm{D}}_{fl}$ which is almost constant. This quantitative difference is due to the $1\ell$ truncation of the fRG [5]. This inconsistency at the two-particle level can be resolved by including multiloop corrections. For this reason, we report also $2\ell$ results[5] that typically represent the largest contribution [4]. Moreover, the $2\ell$ truncation is correct up to $O(U^3)$. In Fig. 15, the difference between $\lambda^{\mathrm{D}}$ and $\lambda^{\mathrm{D}}_{fl}$ is reduced for the $2\ell$ results. Most importantly, also $\lambda^{\mathrm{D}}$ appears to be constant now as well, validating the independence on $U'$ confirming that its dependence on $U'$ is negligible.

It is possible to explain the displayed trends also analytically, by using a "poor-man's" version of the fluctuation diagnostics based on the signs of the different contributions. Starting from Eq. (18) and assuming a positive susceptibility, implies that

$$w^{\mathrm{X}} \leq \mathcal{B}^{\mathrm{X}}, \tag{46}$$

which just confirms that the screened interaction is indeed screened. For $w^{\mathrm{X}} < U^{\mathrm{X}}$ (e.g. if the screening is sufficiently large, or $U'$ sufficiently small) and $\lambda^{\mathrm{X}} \approx 1$, it follows

---

[5]The details on the implementation in the single-boson exchange formulation will be reported in Ref. [88].

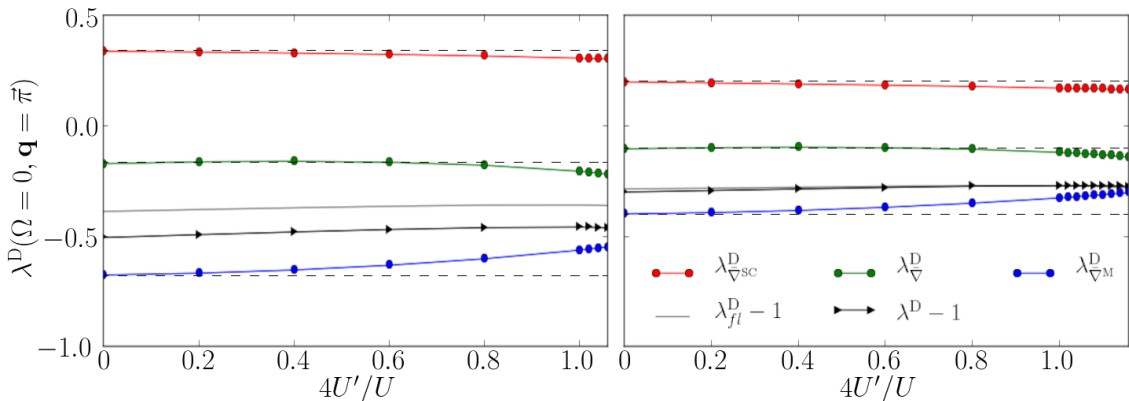

Figure 15: Fluctuation diagnostics of $\lambda^{\mathrm{D}}$ as in Fig. 14 but with additional $2\ell$ corrections.

that $\bar{\nabla} \lesssim 0$. Concentrating on $s$-wave quantities, where secondary fermionic momenta are integrated and the inter-channel projections are simple, we can schematically write $-V^{\mathrm{M}} \sim -V^{\mathrm{D}} \sim V^{\mathrm{SC}} \sim \frac{3}{2}\bar{\nabla}^{\mathrm{M}} - \frac{1}{2}\bar{\nabla}^{\mathrm{D}} - \bar{\nabla}^{\mathrm{SC}}$, leading to

$$\mathcal{I}^{\mathrm{M}} \sim \frac{1}{2}\bar{\nabla}^{\mathrm{M}} - \frac{1}{2}\bar{\nabla}^{\mathrm{D}} - \bar{\nabla}^{\mathrm{SC}} \tag{47a}$$

$$\mathcal{I}^{\mathrm{D}} \sim -\frac{3}{2}\bar{\nabla}^{\mathrm{M}} - \frac{1}{2}\bar{\nabla}^{D} + \nabla^{\mathrm{SC}} \tag{47b}$$

$$\mathcal{I}^{\mathrm{SC}} \sim -\frac{3}{2}\bar{\nabla}^{\mathrm{M}} + \frac{1}{2}\bar{\nabla}^{\mathrm{D}}, \tag{47c}$$

where we used $\mathcal{I}^{\mathrm{X}} = V^{\mathrm{X}} - \nabla^{\mathrm{X}}$. We can then write a sign matrix for the contributions to the vertex $V^{\mathrm{X}}$ from channel $\mathrm{X}'$

$$D_{\mathcal{I}} = \left(D_{\mathcal{I}^{\mathrm{X}}}^{\mathrm{X}'}\right) = \begin{pmatrix} + & - & - \\ - & - & + \\ - & + & 0 \end{pmatrix} \begin{pmatrix} \mathrm{Sgn}\left(\bar{\nabla}^{\mathrm{M}}\right) & 0 & 0 \\ 0 & \mathrm{Sgn}\left(\bar{\nabla}^{\mathrm{D}}\right) & 0 \\ 0 & 0 & \mathrm{Sgn}\left(\bar{\nabla}^{\mathrm{SC}}\right) \end{pmatrix} = \begin{pmatrix} - & + & + \\ + & + & - \\ + & - & 0 \end{pmatrix}, \tag{48}$$

where the rows (from top to bottom) and columns (from left to right) correspond to $\mathrm{M}, \mathrm{D}, \mathrm{SC}$. A positive sign of $(\mathrm{X}, \mathrm{X}')$ means that the contribution from $\nabla^{\mathrm{X}'}$ reinforces $I^{\mathrm{X}}$. On the contrary, a negative sign implies that the contribution from $\nabla^{\mathrm{X}'}$ suppresses $\mathcal{I}^{\mathrm{X}}$. Hence, we call $D_{\mathrm{X}}$ the diagnostic matrix of $\mathcal{I}^{\mathrm{X}}$. Analogous matrices can be formulated for the polarization of Eq. (35), which schematically goes as $P^{\mathrm{X}} \sim \lambda^{\mathrm{X}}\Pi^{\mathrm{X}}$, as well as for the fermion-boson vertex of Eq. (37), as $\lambda^{\mathrm{X}} \sim -\Pi^{\mathrm{X}}\mathcal{I}^{\mathrm{X}}$. The diagnostic matrix for the fermion-boson vertex hence reads

$$D_{\lambda} = -\left(\mathrm{Sgn}\left(\Pi^{\mathrm{X}}\right) \cdot D_{\mathcal{I}^{\mathrm{X}}}^{\mathrm{X}'}\right) = \begin{pmatrix} + & - & - \\ - & - & + \\ - & + & 0 \end{pmatrix} \tag{49}$$

and for the polarization

$$D_P = \left(D_{\lambda^{\mathrm{X}}}^{\mathrm{X}'} \cdot \mathrm{Sgn}\left(\Pi^{\mathrm{X}}\right)\right) = \begin{pmatrix} + & - & - \\ - & - & + \\ - & + & 0 \end{pmatrix}, \tag{50}$$

where the bubbles are considered to be bare susceptibilities, with $\mathrm{Sgn}\left(\Pi^{\mathrm{M}}\right) = \mathrm{Sgn}\left(\Pi^{\mathrm{D}}\right) = \mathrm{Sgn}\left(\Pi^{\mathrm{SC}}\right) = +$. Similarly, we can determine the diagnostic matrix for the vertex. Unlike

$\mathcal{I}^{\mathrm{X}}$, the splitting of the vertex $V^{\mathrm{X}}$ in Eq. (33) contains a contribution from the local part of the bare vertex $U^{\mathrm{X}}$ which does not belong to any channel. The diagnostic matrix thus takes the form

$$D_V^{\nabla} = \left( D_{V^{\mathrm{X}}}^{\nabla^{\mathrm{X}'}} \right) = \begin{pmatrix} - & + & + \\ + & - & - \\ + & - & - \end{pmatrix},$$ 
(51)

with $U^{\mathrm{M}} = -U^{\mathrm{D}} = -U^{\mathrm{SC}} = -U$. Similarly, $D_V^U$ represents the signs of the contributions from the bare interaction

$$D_V^U = \left( D_{V^{\mathrm{X}}}^U \right) = \begin{pmatrix} \mathrm{Sgn}\left(U^{\mathrm{M}}\right) \\ \mathrm{Sgn}\left(U^{\mathrm{D}}\right) \\ \mathrm{Sgn}\left(U^{\mathrm{SC}}\right) \end{pmatrix}.$$ 
(52)

From Eq. (29) follows the diagnostic matrix of the susceptibility to be

$$D_\chi = -\left( \mathrm{Sgn}(\Pi^{\mathrm{X}}) \cdot D_{V^{\mathrm{X}}}^{\nabla^{\mathrm{X}'}} \cdot \mathrm{Sgn}(\Pi^{\mathrm{X}}) \right) = \begin{pmatrix} + & - & - \\ - & + & + \\ - & + & + \end{pmatrix},$$ 
(53)

and

$$D_\chi^U = \begin{pmatrix} -\mathrm{Sgn}\left(U\right) \\ -\mathrm{Sgn}\left(U\right) \\ \mathrm{Sgn}\left(U\right) \end{pmatrix}.$$ 
(54)

These diagnostic matrices describe whether the bosonic fluctuations in a given channel suppress or enhance fluctuations in another channel, but they do not provide any information about the magnitude of these contributions. Although derived by heuristic arguments, we find that the signs perfectly agree with the results of the fluctuation diagnostics displayed in the figures. They also reflect the physical intuition that the magnetic susceptibility is suppressed by density as well as superconducting fluctuations. Conversely, magnetic fluctuations suppress $s$-wave superconductivity, and an $s$-wave superconductivity requires an attractive interaction.

In a similar spirit, we can infer the influence of the bare extended coupling $U'$. Denoting by $\partial\lambda_{\mathrm{X}}^{\mathrm{X}'}$ the derivative with respect to $U'$ of the single-boson contribution to $\lambda^{\mathrm{X}}$ from channel X$'$, we obtain for the linear order in $U'$ (and lowest order in $U$)

$$D_{\partial\lambda^{\mathrm{X}}}^{\mathrm{X}'} = \begin{cases} -D_{\lambda^{\mathrm{X}}}^{\mathrm{D}} \cdot \mathrm{Sgn}\left( \cos\left(q_x^{\mathrm{D}}\right) + \cos\left(q_y^{\mathrm{D}}\right) \right) & \text{if } \mathrm{X}' = \mathrm{D} \\ D_{\lambda^{\mathrm{X}}}^{\mathrm{X}'} D_{\lambda^{\mathrm{X}'}}^{\mathrm{D}} \cdot \mathrm{Sgn}\left( \cos\left(q_x^{\mathrm{D}}\right) + \cos\left(q_y^{\mathrm{D}}\right) \right) \cdot \mathrm{Sgn}\left(U^{\mathrm{X}'}\right) & \text{if } \mathrm{X}' \neq \mathrm{D} \end{cases}$$ 
(55)

evaluated at the ordering vector in the density channel $q^{\mathrm{D}} = (\pi, \pi)$. These expressions stem from the diagrams e) and f) represented in Fig. 16. Compiling the results into a matrix, we obtain

$$D_{\partial\lambda} = \begin{pmatrix} -\mathrm{Sgn}\left(U\right) & - & \mathrm{Sgn}\left(U\right) \\ \mathrm{Sgn}\left(U\right) & - & -\mathrm{Sgn}\left(U\right) \\ -\mathrm{Sgn}\left(U\right) & + & 0 \end{pmatrix}.$$ 
(56)

For a repulsive onsite interaction $\mathrm{Sgn}\left(U\right) = +$, one correctly recovers all the trends of the full calculation. In particular, the second row corresponding to $\lambda^{\mathrm{D}}$, implies that the superconducting and density contributions decrease with $U'$ whereas the magnetic contribution increases. This competition leads to the cancellation responsible of for the

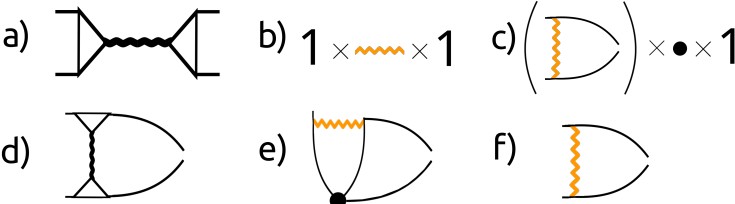

Figure 16: a) Diagrammatic representation of $\nabla^{X'} = \lambda^{X'} w^{X'} \lambda^{X'}$, and b)-c) lowest order contributions in $U'$. d) Contribution from $\nabla^{X'}$ to $\lambda^X \sim \int \mathcal{I}^X \Pi^X$, through a decomposition of $\mathcal{I}^X \sim \sum_{X'} \int \nabla^{X'} \Pi^X$ by Eq. (38), and e)-f) lowest order contributions in $U'$. Since the extended bare coupling $U'$ enters only through the bosonic interaction in the density channel represented by the orange wiggly line, the lowest order contribution occurs in $\nabla^{X'}$. For $X' = D$, the contribution comes from the multiplication with the bare values of $\lambda^D$ and $w^D$ represented by b) of zeroth order in $U$. In this case, the substitution in d) yields a contribution represented by f). For $X' \neq D$, the lowest order contribution comes originates from a density interaction in $\lambda^{X'}$. It is represented by c) and of first order in $U$. In this case, the contribution e) is a result of substituting c) in d).

weak $U'$-dependence and the RPA-like behavior. To what extent this will occur depends on the magnitude of the contributions determined by the full calculation (see Figs. 9 and 12). Another conclusion we can draw is that since there are no negative contributions to $\lambda^{SC}$, they all add up and an RPA description will not apply to $P^{SC}$. Similarly, it holds for $P^M$, where the magnitude of the $s$-wave superconducting contributions are typically small in the repulsive Hubbard model.

## 5    Conclusions and outlook

We have presented an fRG study of the extended Hubbard model including a nearest-neighbor interaction. From the methodological side, we have shown how to extend and apply the concept of the SBE approach to extended interactions by using a modified notion of bare interaction reducibility.

Performing a detailed analysis of different approximations, we have been able to establish an efficient computation scheme that allows us to perform numerically feasible scans of the parameter space. In particular, we determined various physical susceptibilities of the extended Hubbard model at half filling and finite doping. We found that the fermion-boson coupling in the density channel $\lambda^D$ depends only very weakly on $U'$, which leads to an RPA-like behavior for $\chi^D$. Within a fluctuation diagnostics, we were able to illustrate that this observation results from cancellations between the different channel contributions. This trend can be inferred already by a poor-man's version of fluctuation diagnostics restricted to the signs of the contributions (without resorting to the full calculations).

The presented results are obtained by neglecting the flow of the rest function. In the weak-coupling regime, we showed them to have a negligible effect on physical observables. At larger couplings and lower temperatures, where the $d$-wave pairing correlations are expected to become more relevant, the rest function should however be included. In this regime, also the contributions from the mixed bubbles may become relevant. The strong-coupling regime can be accessed within the combination with the DMFT [13], in the so-called DMF$^2$RG [11, 12]. On the methodological side, further developments

include the extension to higher loop orders together and a more accurate treatment of the self-energy flow with the Schwinger-Dyson equation [21, 89, 90] to correctly account for the form factor projections in the TU-fRG. In order to access lower temperatures and consider multiorbital models at a reasonable computational cost, a more efficient and accurate frequency treatment is still needed. Promising candidates are the IR [91, 92] or DLR [93, 94] bases, or the recently introduced Quantic Tensor Trains (QTT) [95] and Tensor Cross Interpolation (TCI) [96].

The SBE for nonlocal interactions developed here can be further extended to long-ranged interactions. It can also be applied to retarded interactions arising e.g. in the Hubbard-Holstein model. Moreover, the SBE formulation offers also the possibility to study mixed boson-fermion systems in presence of additional bosonic fields. Another future direction is the combination with recent advancements [78, 97] in the description of symmetry-broken phases[6].

# Acknowledgements

The authors thank P. Bonetti, A. Kauch, M. Patricolo, L. Philoxene, M. Scherer, A. Toschi, E. van Loon, and D. Vilardi for valuable discussions.

**Author contributions** A. Al-Eryani and S. Heinzelmann contributed equally to this work.

**Funding information** We acknowledge financial support from the Deutsche Forschungsgemeinschaft (DFG) within the research unit FOR 5413/1 (Grant No. 465199066). This research was also funded by the Austrian Science Fund (FWF) 10.55776/I6946. The calculations have been partly performed on the Vienna Scientific Cluster (VSC).

# A  SBE representation in physical channels

Consider the two-particle vertex $V(k_1, k_2, k_3, k_4)$ with $k_1$, $k_3$ corresponding to ingoing legs and $k_2$, $k_4$ to outgoing ones. Assuming translational invariance, its expressions in physical channels are defined by

$$V^{\mathrm{M}}(q, k, k') = -V^{\overline{ph}}(q, k, k') = -V(k, k', k' + q, k + q) \tag{57a}$$

$$V^{\mathrm{D}}(q, k, k') = 2V^{ph}(q, k, k') - V^{\overline{ph}}(q, k, k')$$
$$= 2V(k, k + q, k' + q, k') - V(k, k', k' + q, k + q) \tag{57b}$$

$$V^{\mathrm{SC}}(q, k, k') = V^{pp}(q, k, k') = V(k, k', q - k, q - k'), \tag{57c}$$

where X = M, D, and SC is the magnetic, density, and superconducting channel respectively. As a further step, the SC channel can be split into singlet and triplet being respectively the even and odd component under exchanging the second and the fourth legs

$$V^{\mathrm{sSC}}(q, k, k') = V^{\mathrm{SC}}(q, k, k') + V^{\mathrm{SC}}(q, k, q - k'), \tag{58a}$$

$$V^{\mathrm{tSC}}(q, k, k') = V^{\mathrm{SC}}(q, k, k') - V^{\mathrm{SC}}(q, k, q - k'). \tag{58b}$$

---

[6]See Ref. [2] for an overview.

Note that in diagrammatic channels there is only one (spin-resolved) vertex which can be expressed in the different parametrisations. The vertices in physical channels, however, are projections into a two-particle basis of these channels, and in general do not contain all the information of the vertex.

The SBE representation of the full vertex in physical channels can be derived straightforwardly. In the extended SBE formulation, it holds that $\Phi^{\mathrm{X}} = \nabla^{\mathrm{X}} + M^{\mathrm{X}} - \mathcal{B}^{\mathrm{X}}$, with $\nabla^{\mathrm{X}} = \lambda^{\mathrm{X}} w^{\mathrm{X}} \lambda^{\mathrm{X}}$ for the physical channels. The subtraction of $\mathcal{B}^{\mathrm{X}}$ accounts for the double-counting of the bosonic bare vertex in each $\nabla^{\mathrm{X}}$ that now contains the $\mathcal{B}$-reducible diagrams.

The important claim in the main text is that if we single out the bare interaction contribution from the 2PI part of the vertex

$$V_{2\mathrm{PI}}^{\mathrm{X}} =: V_0^{\mathrm{X}} + \bar{V}_{2\mathrm{PI}}^{\mathrm{X}}, \tag{59}$$

it holds

$$V_{\mathrm{DC}}^{\mathrm{X}} + V_{2\mathrm{PI}} = -2U^{\mathrm{X}} + \bar{V}_{2\mathrm{PI}}^{\mathrm{X}}, \tag{60}$$

where $U^{\mathrm{SC}} = U^{\mathrm{D}} = -U^{\mathrm{M}} = U$. This implies that the expression of the vertex in the parquet approximation ($\bar{V}_{2\mathrm{PI}}^{\mathrm{X}} = 0$) is exactly the one of the purely local Hubbard model also shown in Fig. 1. We will show this for the particle-particle channel for simplicity. Taking care of the channel conversions of the bare interactions (see Appendix C), we have

$$\begin{aligned} V_{\mathrm{DC}}^{pp} + V_0^{pp} &= -\mathcal{B}^{pp} - P^{ph\to pp}\left[\mathcal{B}^{ph}\right] - P^{\overline{ph}\to pp}\left[\mathcal{B}^{\overline{ph}}\right] + \mathcal{B}^{pp} + \mathcal{F}^{pp} \\ &= -P^{ph\to pp}\left[\mathcal{B}^{ph}\right] - P^{\overline{ph}\to pp}\left[\mathcal{B}^{\overline{ph}}\right] + \mathcal{F}^{pp} \\ &= -(\mathcal{F}^{pp} + U) - \mathcal{B}^{pp} + \mathcal{F}^{pp} = -2U. \end{aligned} \tag{61}$$

From the formulation in diagrammatic channels, Eq. (8), we apply Eq. (57) to translate to physical channels. Using Eq. (60), we obtain in the parquet approximation

$$\begin{aligned} V^{\mathrm{M}} =& \nabla^{\mathrm{M}} + \mathcal{I}^{\mathrm{M}} = \nabla^{\mathrm{M}} + M^{\mathrm{M}} + \frac{1}{2}P^{ph\to\overline{ph}}\left[\nabla^{\mathrm{M}} + M^{\mathrm{M}} - \left(\nabla^{\mathrm{D}} + M^{\mathrm{D}}\right)\right] \\ & - P^{pp\to\overline{ph}}\left[\nabla^{\mathrm{SC}} + M^{\mathrm{SC}}\right] - 2U^{\mathrm{M}}, \end{aligned} \tag{62a}$$

$$\begin{aligned} V^{\mathrm{D}} =& \nabla^{\mathrm{D}} + \mathcal{I}^{\mathrm{D}} = \nabla^{\mathrm{D}} + M^{\mathrm{D}} - 2P^{\overline{ph}\to ph}\left[\nabla^{\mathrm{M}} + M^{\mathrm{M}}\right] + 2P^{pp\to ph}\left[\nabla^{\mathrm{SC}} + M^{\mathrm{SC}}\right] \\ & + \frac{1}{2}P^{ph\to\overline{ph}}\left[\nabla^{\mathrm{M}} + M^{\mathrm{M}} - \left(\nabla^{\mathrm{D}} + M^{\mathrm{D}}\right)\right] - P^{pp\to\overline{ph}}\left[\nabla^{\mathrm{SC}} + M^{\mathrm{SC}}\right] - 2U^{\mathrm{D}}, \end{aligned} \tag{62b}$$

$$\begin{aligned} V^{\mathrm{SC}} =& \nabla^{\mathrm{SC}} + \mathcal{I}^{\mathrm{SC}} = \nabla^{\mathrm{SC}} + M^{\mathrm{SC}} - \frac{1}{2}P^{ph\to pp}\left[\nabla^{\mathrm{M}} + M^{\mathrm{M}} - \left(\nabla^{\mathrm{D}} + M^{\mathrm{D}}\right)\right] \\ & - P^{\overline{ph}\to pp}\left[\nabla^{\mathrm{M}} + M^{\mathrm{M}}\right] - 2U^{\mathrm{SC}}, \end{aligned} \tag{62c}$$

where the projection matrices are defined in Ref. [20].

Summarizing, with respect to the SBE implementation of the (local) Hubbard model, the inclusion of the nearest-neighbor interaction of the extended Hubbard model requires to i) change the initial value of $w^{\mathrm{D}} = U \to U + 4V(\cos(q_x) + \cos(q_y))$ and ii) include the non-trivial high-frequency asymptotics of $\lambda^{\mathrm{X}}$ and $M^{\mathrm{X}}$ (see Appendix D). In parameter regimes where the latter may be neglected, the flow equations for $w^{\mathrm{X}}$, $\lambda^{\mathrm{X}}$, and $M^{\mathrm{X}}$, as well as their initial conditions of $\lambda^{\mathrm{X}}$ and $M^{\mathrm{X}}$, are the same.

## B Straightforward SBE extension

The advantages of the SBE implementation as described in Refs. [13, 54] are essentially related to the locality of the bare interaction. For a non-local interaction such as the

Figure 17: Diagrammatic illustration of the straightforward application of the notion of bare interaction reducibility to the extended Hubbard model. The horizontal and vertical lines represent the $\mathbf{q}$ and $\mathbf{k} - \mathbf{k}'$ dependent extended bare interaction respectively.

nearest-neighbor interaction included in the extended Hubbard model, the classification of diagrams according to their bare interaction reducibility can be extended. However, if done naïvely, it undermines the original idea of the SBE as well as the efficiency of its computation.

The strength of the SBE formulation lies in the factorization of the most relevant diagrams into the exchange of a single boson and fermion-boson couplings. A bare nearest-neighbor interaction $V$ that depends also on fermionic momenta defies the purpose of the SBE, since both the 'bosonic' propagators and the fermion-boson couplings acquire an additional fermionic momentum dependence. Fig. 17 shows how the non-local interaction alters the diagrams' structure. As a consequence, the single boson exchange term is no longer a simple product but also contains an internal sum over form factors (see Appendix C for their definition). Moreover, the flow equations of $w^{\mathrm{X}}$ now include an extra form-factor summation. For this naïve extension of the SBE formalism, the flow equations read [20]

$$\dot{W}^{\mathrm{X}}(\Omega, \mathbf{q}, n, n') = \sum_{m, m', l, l', \nu} W^{\mathrm{X}}(\Omega, \mathbf{q}, n, l) \Lambda^{\mathrm{X}}(\Omega, \mathbf{q}, \nu, l, m) \dot{\Pi}^{\mathrm{X}}(\Omega, \mathbf{q}, \nu, m, m')$$
$$\times \bar{\Lambda}^{\mathrm{X}}(\Omega, \mathbf{q}, \nu, m', l') W^{\mathrm{X}}(\Omega, \mathbf{q}, l', n') \tag{63a}$$

$$\dot{\Lambda}^{\mathrm{X}}(\Omega, \mathbf{q}, \nu, n, n') = \sum_{m, m', \nu'} \Lambda^{\mathrm{X}}(\Omega, \mathbf{q}, \nu', n, m) \dot{\Pi}^{\mathrm{X}}(\Omega, \mathbf{q}, \nu', m, m') \mathbf{I}^{\mathrm{X}}(\Omega, \mathbf{q}, \nu', m', \nu, n'), \tag{63b}$$

where we used capital letters and boldface $\mathbf{I}$ to distinguish those quantities from the ones in the main text. The flow equation for the rest function is unchanged. It still holds $\Lambda^{\mathrm{X}}(\Omega, \mathbf{q}, \nu, n, \nu', n') = \bar{\Lambda}^{\mathrm{X}}(\Omega, \mathbf{q}, \nu, n', \nu', n)$, which can be exploited to reduce the computational effort. Note that the two form-factor dependencies of the fermion-boson couplings are not the same in nature and $\Lambda^{\mathrm{X}}(\Omega, \mathbf{q}, \nu, n, \nu', n') \neq \Lambda^{\mathrm{X}}(\Omega, \mathbf{q}, \nu, n', \nu', n)$: The dependence on the incoming form factor does not change with a non-local interaction, while the outgoing form factor couples only to the bare interaction. Therefore, the first shell of form factors accounts for the full fermionic momentum dependence. The corresponding (non-zero) initial conditions are

$$W^{\mathrm{X,init}}(\Omega, \mathbf{q}, n, n') = V_0^{\mathrm{X}}(\mathbf{q}, n, n') \tag{64a}$$

$$\Lambda^{\mathrm{X,init}}(\Omega, \mathbf{q}, \nu, n, n') = 1 \quad \text{for} \quad n = n' < 5, \tag{64b}$$

see Appendix C for the explicit form-factor dependence of $V_0^X$. At the same time, the inclusion of the full first shell of form factors, together with the additional internal summations, renders this approach numerically unfeasible. From this point of view, the SBE formalism may not appear to be a natural framework to treat non-local interactions, in contrast to the conventional fermionic fRG implementation not plagued by this problem[7].

In the following, we will establish the relation between the quantities expressed in terms of $V_0$-reducibility to those in terms of $\mathcal{B}$-reducibility introduced in the main text. For brevity, we denote the integration of the fermionic arguments with a circle. For the bosonic propagators, we have in the diagrammatic channels

$$
\begin{aligned}
W_r &= V_0^r - V_0^r \circ \chi^r \circ V_0^r \\
&= w_r + \mathcal{F}_r - \mathcal{B}_r \circ \chi_r \circ \mathcal{F}_r - \mathcal{F}_r \circ \chi_r \circ \mathcal{F}_r - \mathcal{F}_r \circ \chi_r \circ \mathcal{B}_r,
\end{aligned}
\tag{65}
$$

where we used $V_0^r = \mathcal{B}_r + \mathcal{F}_r$ and $w_r = \mathcal{B}_r - \mathcal{B}_r \circ \chi^r \circ \mathcal{B}_r$ (or, equivalently, Eq. (18)) to obtain the second line. Alternatively, one has

$$
W^r = V_0^r - V_0^r \circ \left[ \mathcal{B}_r^{-1} \circ (\mathcal{B}_r - w_r) \circ \mathcal{B}_r^{-1} \right] \circ V_0^r.
\tag{66}
$$

The situation for the fermion-boson coupling $\Lambda_r$ is less straightforward. In the definition (37), the $\mathcal{B}$-irreducible vertex $\mathcal{I}_r$ is replaced by the $V_0$-irreducible vertex $\mathbf{I}_r$. Splitting $V_0^r$ yields diagrams in $\mathcal{I}_r$ that are not contained in $\mathbf{I}_r$: the $\mathcal{F}_r$-reducible ones. To obtain $\mathcal{I}_r$ from $\mathbf{I}_r$, one thus has to glue all diagrams in $\mathbf{I}_r$ with $\Pi_r \circ \mathcal{F}_r \circ \Pi_r$. We have

$$
\begin{aligned}
\mathcal{I}_r = {} & \mathbf{I}_r + \sum_{n=1}^{\infty} (1 + \mathbf{I}_r \circ \Pi_r) \circ (\mathcal{F}_r)^n \circ (1 + \Pi_r \circ \mathbf{I}_r) \\
& + \sum_{n=1, m=1}^{\infty} (1 + \mathbf{I}_r \circ \Pi_r) \circ (\mathcal{F}_r)^n \circ \Pi_r \circ \mathbf{I}_r \circ \Pi_r \circ (\mathcal{F}_r)^m \circ (1 + \Pi_r \circ \mathbf{I}_r) + \cdots,
\end{aligned}
\tag{67}
$$

where $(\mathcal{F}_r)^n := \mathcal{F}_r (\circ \Pi_r \circ \mathcal{F}_r)^{n-1}$. Summing up the geometric series, we obtain

$$
\mathcal{I}_r = \mathbf{I}_r + (1 + \mathbf{I}_r \circ \Pi_r) \circ \mathbf{F}_r \circ (1 - \Pi_r \circ \mathbf{I}_r \circ \Pi_r \circ \mathbf{F}_r)^{-1} \circ (1 + \Pi_r \circ \mathbf{I}_r),
\tag{68}
$$

with $\mathbf{F}_r := \mathcal{F}_r \circ (1 - \Pi_r \circ \mathcal{F}_r)^{-1}$. Through the relation between $\mathbf{I}_r$ and $\mathcal{I}_r$, one can relate also the rest functions. Denoting the full $V_0$-reducible rest function by $\mathbf{M}_r$, one has that

$$
S_r := \mathbf{I}_r - \mathbf{M}_r = \mathcal{I}_r - M_r - \mathcal{F}_r,
\tag{69}
$$

with $S_r$ being the $GG$-irreducible vertex in channel $r$ minus the bare interaction. Finally, we mention that if the generalised SBE quantities $M_r$ and $\mathcal{I}_r$ are known, it is possible to reconstruct $\Lambda_r$ directly using $S_r$ and the Bethe-Salpeter equation given by [20]

$$
\Lambda_r = 1 + S_r \circ \Pi_r \circ \Lambda_r.
\tag{70}
$$

Equations (66) and (68)-(70) establish the relations connecting both decomposition schemes. Note that when $V_0^r = \mathcal{B}_r$ (or equivalently $\mathcal{F}_r = 0$), they become trivial.

---

[7]The Wentzell asymptotic functions $K_1$ and $K_2$ do acquire an additional form-factor arguments, but the flow equations are the same.

## C Form-factor expansion of the bare interaction

The onsite and nearest-neighbor bond form factors in momentum space are given by

$$f_0(\mathbf{k}) = 1, \qquad\qquad\qquad\qquad\qquad \text{onsite} \qquad (71a)$$

$$f_1(\mathbf{k}) = e^{i\mathbf{k}_x}, f_2(\mathbf{k}) = e^{i\mathbf{k}_y}, f_3(\mathbf{k}) = e^{-i\mathbf{k}_x}, f_4(\mathbf{k}) = e^{-i\mathbf{k}_y}, \qquad \text{1st shell} \qquad (71b)$$

The $s$-wave and $d$-wave form factors used here can then be defined as

$$f_{s-\text{wave}} = f_0, \quad f_{d-\text{wave}} = \frac{1}{2}(f_1 + f_3) - \frac{1}{2}(f_2 + f_4) = \cos \mathbf{k}_x - \cos \mathbf{k}_y. \qquad (72)$$

In physical channels, the form-factor expansion of the bare interactions reads

$$V_0^{\text{X}}(\mathbf{q}, n, n') = \mathcal{B}^{\text{X}}(\mathbf{q}) + \mathcal{F}^{\text{X}}(n, n'). \qquad (73)$$

Considering bond form factors explicitly gives

$$\mathcal{F}^{\text{M}}(n, n') = -2U'(\delta_{n,1}\delta_{n',1} + \delta_{n,2}\delta_{n',2} + \delta_{n,3}\delta_{n',3} + \delta_{n,4}\delta_{n',4}) \qquad (74a)$$

$$\mathcal{F}^{\text{D}}(n, n') = -2U'(\delta_{n,1}\delta_{n',1} + \delta_{n,2}\delta_{n',2} + \delta_{n,3}\delta_{n',3} + \delta_{n,4}\delta_{n',4}) \qquad (74b)$$

$$\mathcal{F}^{\text{SC}}(n, n') = 2U'(\delta_{n,1}\delta_{n',1} + \delta_{n,2}\delta_{n',2} + \delta_{n,3}\delta_{n',3} + \delta_{n,4}\delta_{n',4}), \qquad (74c)$$

whereas $\mathcal{F}$ vanishes on both the $s-$wave and the $d-$wave form factors. Despite the fact that the interaction may exhibit a different form in the different channels, there is only a single bare interaction. This is highlighted by the relation $V_0^r(\mathbf{q}, n, n') = P^{r' \to r}[V_0^{r'}](\mathbf{q}, n, n')$ for the diagrammatic channels. Here $P^{r' \to r}$ is a matrix in form factor space which implements[8] the reparametrizations of the secondary momenta in translating between the different diagrammatic channels shown in Eq. 57. The splitting into bosonic/horizontal and fermionic/vertical parts complicates the picture since $\mathcal{B}^r \neq P^{r' \to r}[\mathcal{B}^{r'}]$ and $\mathcal{F}^r \neq P^{r' \to r}[\mathcal{F}^{r'}]$ in general. Translating between the $\overline{ph}$- and $pp$-channel has no effect on $\mathcal{B}$ or $\mathcal{F}$. In contrast, the translation to and from the $ph$-channel turns $\mathcal{B}$ to $\mathcal{F}$ and vice versa. For the diagrammatic channels (analogous relations for the physical channels may be easily derived) holds

$$\mathcal{B}^{ph}(\mathbf{q}, n, n') = P^{\overline{ph}/pp \to ph}\left[\mathcal{F}^{\overline{ph}/pp}\right](\mathbf{q}, n, n') + U \qquad (75a)$$

$$\mathcal{B}^{\overline{ph}/pp}(\mathbf{q}, n, n') = P^{ph \to \overline{ph}/pp}\left[\mathcal{F}^{ph}\right](\mathbf{q}, n, n') + U, \qquad (75b)$$

and, respectively,

$$\mathcal{F}^{ph}(\mathbf{q}, n, n') = P^{\overline{ph}/pp \to ph}\left[\mathcal{B}^{\overline{ph}/pp}\right](\mathbf{q}, n, n') - U \qquad (76a)$$

$$\mathcal{F}^{\overline{ph}/pp}(\mathbf{q}, n, n') = P^{ph \to \overline{ph}/pp}\left[\mathcal{B}^{ph}\right](\mathbf{q}, n, n') - U. \qquad (76b)$$

Note that in switching form $\mathcal{B}$ to $\mathcal{F}$ and back, one has to be careful to avoid double counting of the local bare interaction.

## D Extended asymptotics for $\lambda$ and $M$

Within the generalized SBE decomposition presented here, the bosonic propagator $w^{\text{X}}$ absorbs the bosonic dependence of the extended bare interaction. The residual fermionic

---

[8]The explicit expressions for these matrices can be found in e.g. Refs. [20, 98].

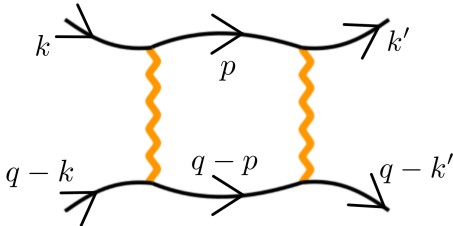

Figure 18: An example contribution to $M^{\mathrm{SC}}$ of the extended Hubbard model which would does not decay to zero as $\nu, \nu' \to \infty$.

"vertical" momentum-dependent bare interaction will however contribute to diagrams of $\lambda^{\mathrm{X}}$ and $M^{\mathrm{X}}$, leading to non-trivial frequency asymptotics. Consider for example the diagram in Fig. 18, representing a contribution to $M^{\mathrm{SC}}$ (see Fig. 3). It reads

$$\sum_{\mathbf{p}, i\nu''} \mathcal{B}^{\mathrm{D}}(\mathbf{p} - \mathbf{k}) G(i\nu'', \mathbf{p}) G(i\Omega - i\nu'', \mathbf{q} - \mathbf{p}) \mathcal{B}^{\mathrm{D}}(\mathbf{k}' - \mathbf{p}) \in M^{\mathrm{SC}}(q, k, k') \qquad (77)$$

and depends on all the three momenta $\mathbf{q}, \mathbf{k}$ and $\mathbf{k}'$, but only on the frequency $\Omega$. Thus, this diagram survives in the limit $\nu, \nu' \to \infty$ and is part of $M_2^{\mathrm{X}}$, defined in the main text by Eq. (24b).

We have verified that these additional contributions to the asymptotics are negligible in practice. The influence of both the asymptotic fermion-boson vertex denoted by $\lambda^{\mathrm{asympt}}$ and the asymptotic rest functions $M_1^{\mathrm{X}}$ and $M_2^{\mathrm{X}}$ introduced in analogy to the classification of Ref. [17] is reported in Fig. 19, for the same parameters as in Fig. 5. For $w^{\mathrm{X}}$, we find that including $M_1^{\mathrm{X}}$ and $M_2^{\mathrm{X}}$ leads to corrections < 1 %. As for the fermion-boson vertex asymptotics $\lambda^{\mathrm{asympt}}$, the relative differences are below 1% for the bosonic propagators and even smaller (< 0.1%) for $\lambda^{\mathrm{X}}(n = s\text{-wave})$. For the non-local form factors of the first shell, the relative difference in the fermion-boson coupling is much larger, up to 15% (see Ref. [20]). However, the absolute values of the corresponding components of $\lambda^{\mathrm{X}}$ are negligible in comparison to the local one and unless one is interested in their study, they can be safely omitted.

For the considered parameter regime, the non-trivial frequency asymptotics does not affect the presented results.

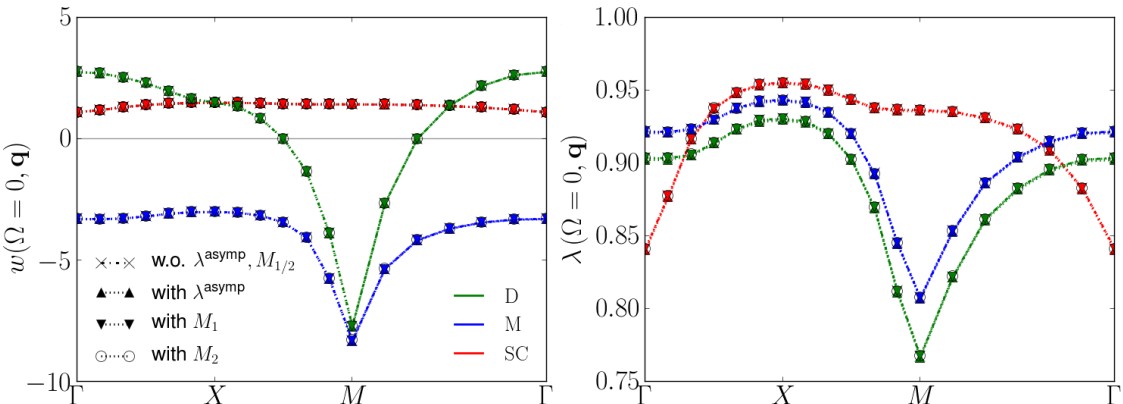

Figure 19: Bosonic propagator $w^{\mathrm{X}}$ and fermion-boson vertex $\lambda^{\mathrm{X}}$ as obtained from the calculation with and without high-frequency asymptotics of $\lambda^{\mathrm{asympt}}$ as well as with and without rest function asymptotics, for the same parameters as in Fig. 5.

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
