# Peer review of "Screening and effective RPA-like charge susceptibility in the extended Hubbard model"

_SciPost Physics_

## Round 1 · Referee Report · Daniel Rohe (Referee 2) · 2025-1-15

Report
Preface:
I consider the manuscript to be eligible for publication in SciPost, provided that the comments below are considered within the process of a revision.
My feedback is phrased in more general terms first, and complemented by page-specific comments after. The page-related items are tagged as either "optional" or "recommended", to indicate the relevance I personally would assign to them. I hope that the overall set of comments will serve well to help improve certain aspects of the work in a second iteration.
Regarding item 6 of the relevant acceptance criteria, I am somewhat uncertain to which level they are required on behalf of SciPost - see comment below.
* * *
Summary:
The authors report on the application of a recently developed scheme for numerical functional renormalisation group (fRG) computations to the extended Hubbard model. Here, the latter contains a nearest-neighbour density-density-interaction in addition to the local interaction. The so-called single-boson exchange (SBE) representation of the Fermion-Fermion interaction provides some inherent advantages, in conjunction with fRG mainly being an increased computational efficiency on the numerical side, and a way to avoid divergencies otherwise present in a 2-particle-irreducible formalism on the formal side.
The core progress offered in this work consists in a careful formal as well as practical extension of the SBE-fRG formalism to nonlocal bare interactions, which is non-trivial and requires an explicit and additional route to suitably parametrise the two-particle interaction and bosonic representations thereof. Along with the formal development, selected numerical results are reported and related to other approaches. Numerical findings are further complemented by a high-level fluctuation analysis which provides a rough plausibility check for general properties of certain quantities during the fRG flow.
With respect to the expectations of SciPost, this work does in my opinion to a sufficient extent
i) "provide a novel and synergetic link between different research areas", namely by connecting and combining different recent methods and approaches in correlated condensed matter theory that are a priori of different origin, and subsequently fusing them in a numerical framework, which is then implemented and applied to some example cases,
and
ii) "open a new pathway in an existing or a new research direction, with clear potential for multi-pronged follow-up work", due to the increased numerical efficiency, most importantly due to a more favourable scaling property of the size of the resulting numerical problem compared to various other fRG approaches, while all the same maintaining other formal advantage of the SBE-fRG formalism, and also keeping the option to treat stronger bare interactions by combining the fRG treatment with an "initialisation" of the fRG flow by means of other methods, namely and most importantly DMFT.
In my opinion, the manuscript generally matches the acceptance criteria, subject to specific remarks given in various comments below.
Regarding item 6 in the list of these acceptance criteria, on providing "all reproducibility-enabling resources": I am not quite sure to what degree this is required at an obligatory level on behalf of the journal. In my personal view, the work does provide within a reasonable scope enough information to be able to reproduce the computations to a sufficient degree, by means of equations and information on numerical implementations. While it would be an additional and welcome step to publish the actual source code and raw data, etc., which I do of course encourage, I personally would not require it as an obligatory measure. There can be imho good and fair reasons to not make a code publically available.
In any case, I would appreciate some clarification from the editorial side on this. I feel that a firm and proper judgement on this criterion is beyond the scope of my personal feedback on this manuscript/work. I would also kindly like to ask the authors to provide their view on this in their reply to this report, see also one of the comments below.
* * *
----- Report section -----
* * *
I - General questions and comments
* * *
A - Physical and methodical aspects
i) The extension of the model here stems from a density-density-term, translating into a leading relevance for the density-density-related channel within the formalism. Yet, the model can also be extended in other ways, also in combinations.
-> What would happen if a spin-spin-term was also added, as e.g. in DOI 10.1103/PhysRevB.67.125104? Can the SBE be adapted to this case, too? If yes, would there be prospects and expectations for such an investigation?
ii) I found it a bit involved and difficult to figure out from the text which approximations are made a priori on general mathematical grounds, which stem from example data, and which stem from physical arguments. I would thus find it helpful if the approximations made to arrive at the final equations that are used in the numerical implementation could be summarised, e.g. as a brief tabular overview, including in particular some (short) comments on the arguments for and justification of the respective approximations. I feel such an overview could help the reader to better be able to follow the route from the intially exact formulation and parametrisation to the computationally efficient implementation of the SBE.
iii) Concerning the matter of divergences in the two-particle irreducible (2PI) vertex which can be avoided in the SBE, in contrast to other methods:
Being a non-expert in this area, but having noticed intense activity in recent years on the matter, I wonder if some more information can be given on (tentative) physical interpretations. I.e., do these divergences
a) in any way a transition or signal a break-down of the normal phase, thus carrying a physical meaning? If that is the case, avoiding them might also mean "missing" some physics
or
b) is it clear that these divergences are purely technical obstacles appearing in the 2PI formalism, which can be certain to not signal true phase transistions, but at most e.g. a cross-over into a regime where the physics of the bare metallic model is a truly bad starting point in the perturbative sense?
C.f. also the discussions in e.g. Ref. 55 and Ref. 60 , Ref. 64 , Ref. 65.
Maybe it is possible to add a brief comment on this.
iv) If I understand correctly, the multiboson/2PI contributions which are neglected in the SBE-fRG implementation do become relevant in pseudo-critical regions, i.e. in particular near "one-loop phase transitions". Of course, also standard perturbative fRG is limited, and eventually breaks down, close to such a critical point.
-> Given this qualitatively phrased restriction of scope in applicability, in how far can the SBE-fRG be as valid for the purpose of computing e.g. phase diagrams as more conventional fRG schemes are? In other words: If e.g. a phase transition was analysed via divergent effective interactions and/or susceptibilities, is the SBE-fRG more restricted that conventional schemes, or is it just as suitable/restricted when entering a region of pseudo-criticality compared to other fRG schemes?
C.f. also some suggestions of additional results to compare against previous (fRG) works as given below.
* * *
B - Computational and practical aspects
i) Since computational efficiency is one of the central properties of the method, serving as a motivation and objective, I would suggest to present this aspect in a somewhat more explicit and quantitative way. Orders of magnitude and examples would suffice, imho, both concerning runtimes and resources used.
In that light, I would suggest to include information such as e.g.:
a) How many nodes/cores on which architecture are used for the computation?
b) How long are the runtimes?
c) How many degrees of freedom are actually treated in the final ODE, and how does this constitute an advantage over previous approaches?
d) What is the level of parallelism that the implementation prifts from, and how does the code scale?
e) What are the computational limits in the current setup, in terms of e.g. resolution, reachable points in parameter space, speed-up/runtimes, parameter scans, etc., and in how far can/do these limits extend the limits imposed by previous approaches?
This might best be suited for an additional appendix. For this discussion, also the following references (and maybe others) dealing with computational aspects in fRG applications might be relevant or helpful:
- Reference [19] in the mansucript
- DOI: 10.21468/SciPostPhysCodeb.26
- DOI: 10.1016/j.cpc.2016.05.024
ii) I would appreciate a comment of the authors regarding item 6 of the acceptance criteria - see above.
* * *
C - Results
i) Several statements are made about the accuracy and reliability of the SME method. I would find it helpful to better specify and quantifiy the terms "accurate" and "reliable" in this context: Do they e.g. refer to comparisons with existing (an possibly exact) results, to internal error and consistency checks, or to something else?
ii) I would encourage the authors to devote some more content to comparisons of the obtained results to existing results, or to emphasise when results are new and elaborate on this.
iii) Given that the SBE-based fRG is quite versatile and efficient, I feel that the result section leaves some room for additional representative parameter sets to be investigated, or for the representation of additional data (e.g. for the self-energy) which might already be available but has not been displayed.
As a function of time and resources that can be spared, I could think of the following cases/questions as examples. Maybe ONE or TWO of such results can be added and included (The list is NOT meant as a requirement, but shall provide some ideas only):
a) Include a J-term in the bare model (nearest-neighbour spin-spin-interaction)
c.f. 10.1103/PhysRevB.67.125104
I would expect this to promote the spin channel to the same zeroth-order level as the charge channel in terms of the q-dependence, thus providing also an example of an additional methodical extension.
* * *
b) Add a point or two in parameter space for negative U'
c.f. e.g. 10.1038/s42005-022-01030-x and 10.21468/SciPostPhys.17.2.054 (1d-case) and 10.1103/PhysRevB.67.125104 (2d case) and compare results.
A negative U' can e.g. potentially favour/trigger/enhance superconductivity of various kinds.
* * *
c) Look at the case of attractive U and repulsive V -> c.f. reference 23:
Does a positive U' cut off the superconducting transition also in the SBE-fRG -> Fig. 7a) in Ref. 23 ?
* * *
d) Single out a case where d-wave SC actually IS dominant, in case that exists in SBE-fRG. If it is not found, this would be even more interesting.
* * *
e) Can strong charge fluctuations induce a pseudogap in SBE-fRG? C.f. arxiv 2411.11661 , Ref. 24 (Fig. 8), Ref. 25.
* * *
f) Can the SBE-fRG (with DMFT-init) be applied to and confirm the transition from Mott physics to a charge-ordered insulator - c.f. Ref 25?
* * *
g) Self-energy and Mott physics: What happens in SBE-fRG (with DMFT-init) at half filling and moderate to large frustration, keeping e.g. t'=-0.2 and maybe increasing it to t'=-0.3?
* * *
h) Does the SBE capture the AFM-SC-FM transition at van-Hove filling as a function of t' at small enough temperatures?
c.f. e.g. DOI 10.1103/PhysRevLett.87.187004
* * *
i) Provide a comparison to Fig. 1 in Ref 24 -> phase diagram
* * *
j) Provide a comparison to Fig. 5 in Ref 35 -> are such low temperatures feasible?
* * *
k) Provide and discuss self-energy data and compare with e.g. Ref 38 Fig. 2, Fig. 5
* * *
l) In relation to Ref. 42, Fig. 12: There, d-wave sc only appears for negative V at all. fRG generates d-wave at V=0. Can SBE-fRG shed some additional light on this?
* * *
m) Compare d-wave interactions from SBE-fRG with Ref. 44
* * *
n) Can the question of potential triplett pairing be addressed by the method, c.f. Ref. 51 (different lattice symmetry, but statement that triplet pairing relies mainly on nonlocal interactions) , and c.f. DOI 10.1038/s42005-022-01030-x , in conjunction with an attractive V, and also Désoppi (1d-models)
* * *
D - Conclusion
i) I suggest to divide the conclusion more clearly into
a) a summary of the work and its relation to previous works, and
b) a forward-looking part for prospective works, which are made accessible via the very method of SBE-fRG.
ii) I also suggest to add a comparison to previous works, emphasising on
a) previously existing results that were reproduced by the method, and if this can serve as a benchmark for the method,
b) potential differences to previous results (where/if applicable),
c) results that are new and their implications.
* * *
E - Other
i) For some references pointing to preprints, the hyperlinks behind the identifier failed for me.
ii) Ref [20] is apparently not yet publically available. I suggest to add a comment to the citation such as "to be published".
iii) References [80] and [81] seem to be identical.
iv) The following additional references might be useful/relevant/related (see also below for some of them):
- DOI 10.1103/PhysRevB.102.195109
- DOI 10.1088/1742-6596/391/1/012148
- DOI 10.1103/PhysRevB.67.125104
- DOI 10.1038/s42005-022-01030-x
====================
II - Questions and comments by page
* * *
Page 2 - optional:
quote: "... and are not trivially generalized to nonlocal interactions"
-> This sounds potentially ambiguous to me. I guess what is meant is: "It is not trivial to extend the method to the case of nonlocal interactions", which is also evident from the sentence that follows. Maybe this can be phrased a little clearer (then again, this might only be my personal perception)
* * *
Page 2 - recommendation:
quote: ".. providing a numerically feasible fRG-based computation scheme for treating a nearest-neighbor interaction."
-> This sounds as if it may not yet have been possible to treat the/an extended Hubbard model in fRG at all. In some fRG schemes this has of course been done to a certain extent, as stated on page 3 and cited in the manuscript in e.g. ref. [44-46], see also for example DOI 10.1103/PhysRevB.67.125104 .
I recommend to add a short overview of existing fRG works on the extended Hubbard model and to specify more explicitly the motivation and benefit for developing and employing the SBE framework in addition to these existing fRG schemes and works. I would suggest to add this to the paragraph on page 3 that starts with "With the fRG" and already partially gives such an overview.
I would also kindly suggest to account for these additional references in that context:
- DOI 10.1103/PhysRevB.67.125104 (2d case)
- DOI 10.21468/SciPostPhys.17.2.054 (1d case)
* * *
Page 3 - optional:
quote: "... low temperature regime"
-> Given the temperatures chosen later in the results, I would find it helpful to here mention what "low" actually means. Temperatures at and above T=0.2 are imho already outside a "low temperature regime".
* * *
Page 3 - recommendation:
quote: "Recently, the SBE ..."
I find the subsequent paragraph somewhat difficult to follow:
i) It is stated that "At weak coupling, the multiboson processes are irrelevant". Whether in turn they are relevant at strong coupling, and if yes in how far, is however not mentioned. Instead, in the sentence that immediately follows, statements about "advantages" of the SBE also at strong coupling are made. These are two entirely different aspects, though, namely validity and practicality, which can - and often enough do - turn out to be partially incompatible with each other.
ii) It is stated that "no diagrammatic element of the SBE decomposition of the vertex function displays [53, 71] the non-perturbative divergencies that plague their parquet counterparts."
By all given references on the issue of divergences in the 2PI-vertex, I am still not fully aware whether these divergencies are purely technical/artificial and not related to physics, or whether they can/may imply substantial changes or even phase transistions. There is ongoing work in this direction, and I am not an expert. Yet, given this statement I am left uncertain whether avoiding these divergencies means that a method is better suitable, or whether it may mean that some physics is possibly NOT captured BECAUSE these divergencies are circumvented. ( C.f. also general comments above)
-> I suggest to split and slightly extend this paragraph, in order to
i) disentangle more clearly aspects of relevance and irrelevance of certain contributions, here specifically the multiboson terms, at weak and strong bare interaction, from practical advantages and disadvantages of the numerical implementation of the SBE-fRG method.
ii) (if possible) explain a little more clearly the scope of of validity of the SBE with respect to avoiding divergencies in the relevant two-particle irreducible quantities.
* * *
Page 4 - optional:
quote "Diagrams that can not, are U-irreducible and correspond to multiboson processes."
This statement left me contemplating for a while: Can all these contributions really be called "multi-boson" in the physical sense? To the best of my understanding, there are diagrams and processes in this set which I would not expect to be classified in terms of "many bosons", even in the wider and typical sense of a "boson" as a collective two-fermion excitation. Thus, I wonder if this "label" is valid in physical terms.
-> I assume this classification is by now inherent to the SBE (and in more general terms), but I would appreciate a brief note on this, if possible.
* * *
Page 5 - recommendation:
Eq. 9c): I don't quite undestand what limit is actually meant here. It would help if this could be made clearer.
* * *
Page 7 - recommendation:
quote: "... the nearest-neighbor interaction enhances the charge-density wave fluctuations fully encoded in wD. The influence of U′ on the other objects is expected to be of secondary importance."
If I understand correctly, this is an a priori statement of expectation. I do not fully understand if the point that is made means that this is an assumption which is USED to develop the formal matters which follow, or whether the SBE remains unbiased in this respect and can VERIFY this expectation.
-> I would find it helpful if this could be made clearer.
* * *
Page 7 - optional:
Fig 3: I do not fully understand what these diagrams represent exactly, and the specific graphical representation feels a little unfamiliar to me.
-> I think I get the point, yet I feel it would help to offer this diagrammatic view also for the local interaction, to make those differences obvious which are relevant for the purpose of this work. C.f. also comment below on the non-trivial frequency dependence that arises, as far as I understand due to this difference.
* * *
Page 8 / Appendix D - optional:
quote: "...fermion-boson couplings and the rest functions develop non-trivial high-frequency asymptotics..."
-> Refer to Appendix D here already (As far as I understand, that this the very matter.)
It is probably very simple, but I am somehwat stuck at this point: As far as I understand, this non-trivial frequency asymptotics arises due to the additional splitting in the respective channels in bosonic and fermionic parts, with a bosonic q-dependence appearing in the charge channel that then couples to the fermionic parts in the other channels, as depicted in Fig. 18 and stated in Eq. (77). What I do not quite understand: Why does this not happen in the purely local case? What happens in the standard local case and without this additional splitting, to the diagram shown in Fig. 18? I note that for B=0 the contribution vanishes, but without the splitting I would expect the diagram to still contribute, on the basis of V(k...) rather than on the basis of B + F.
Maybe this can be clarified by an additional sentence, equation and/or diagram in Appendix D, where the local case could be presented for direct comparison. C.f. previous comment.
* * *
Page 9 - optional:
Eq. 20a and 20b: The fRG equations include self-energy feedback, on the "standard" level based on (GS)-loops, as opposed to the level of the often used so-called "Katanin replacement", i.e. where the full loop (GG) is subject to differentiation with respect to the regulator.
-> It would be helpful if the reason for this choice was briefly mentioned (maybe I also overlooked it).
* * *
Page 10 - recommendation:
quote: "Moreover, since the additional momentum dependence stems from the extended interaction, it will involve only the bond form-factor indices corresponding to the range of the interaction."
Which interaction is meant here, and which range? The effective interaction may (and I expect does) generally build up longer ranged components during the flow, even more so when a critical region is approached and long-range correlations set in.
-> I suggest to clarify this point, and its relevance for the choices and properties of the practical implementation.
* * *
Page 10 - recommendation:
quote: "we use a smooth frequency cutoff"
-> I may have missed it, but I did not find information on the initial value of \Lambda at the start of the flow, i.e. \Lambda_0. Is it truly \Lambda_0 = \infty? In that case, I would assume the technical parametrisation to be made with a different, auxiliary flow parameter.
On the other hand, if it is not \Lambda_0=\infty, but e.g. a large but finite value, I would appreciate some more specific information on and arguments for the initial conditions that are used.
* * *
Page 14 - optional:
quote: "We first observe that the so-called mixed bubbles [...] are in general significantly smaller than the ones diagonal in form factors"
Is this an observation that is made in the numerical data? Then it could maybe be shown, e.g. by plotting or at least giving some numbers of the magnitude of the off-diagonal contributions with respect to the diagonal ones.
If, in contrast, the observation is made a priori on the basis of terms and equations, I would suggest to briefly state this.
* * *
Page 14 and 16 - recommendation:
As an addition to Fig. 5 I would find it helpful to also illustrate the fRG flow in the traditional, common way of plotting the corresponding effective interaction as a function of the flow parameter, in the leading sectors. This could give an idea about the magnitude of the effective interaction, which is often/typically compared to a stopping criterion in case a divergence develops before all degrees of freedom are "integrated out".
Maybe this could be done by adding a couple of extra boxes to the plot and showing the flow of the largest couplings (presumably at \omega=0) in e.g. the charge, spin and pairing channels.
The same appplies to Fig. 6, where it is illustrated that with smaller temperatures the susceptibilities start to peak and show early signs of divergences. Also here I would suggest to add conventional plots which show the flow of the effective interaction, since this quantity usually also increases when strong/long-ranged correlations develop.
* * *
Page 15 - recommendation:
quote: "the run times for the current implementation including mixed bubbles are numerically also not feasible"
c.f. main comments on computational aspects -> more specific information on run times would be valuable, to corroborate and highlight the advantage of an increased numerical efficiency that the method provides.
* * *
Page 15 - recommendation:
quote: "we thereby assume that the approximations [...] do not alter its validity ..."
Given that Fig. 5 shows results for an elevated temperature, leading to behaviour which is more on the regular side, this statement does not quite convince me, also given that it is not justified by additional arguments or conjectures. I encourage to add arguments why this assumption is made, and even more importantly why it can be expected to remain valid also when correlations become more relevant, in light of the smaller temperatures that are chosen for other results that follow later.
Alternatively, from another prespective, it might be useful to add a comment under which cases these contribution COULD actually be relevant and shall NOT be discarded, and how these cases differ from the cases looked at in this work.
* * *
Page 15 - recommendation:
quote: "... the effect of a form-factor truncation turns out to be very small and a quantitatively accurate description ..."
Also here I encourage to outline more explicitly and precisely arguments on the expected validity and limits of this statement, even more so since it made in conjuction with the notion of "accuracy".
* * *
page 15 - recommendation:
quote: "...which appear to be accurately described by a calculation neglecting all nonlocal form factors."
Similarly, this statement leaves me in a bit of an uncertain state. "Appears to be" is a comparatively weak and unspecific statement, yet it is made in conjunction with "accuracy", being a claim of a stronger kind. If possible, I would favour clearer and more robust statements in (semi-)quantitative terms concerning both aspects: The meaning of "appear to be" as well as the meaning of "accurately described".
* * *
page 15 - recommendation:
quote: "It highlights how the SBE decomposition serves not only to reduce the numerical effort but also to gain a deeper understanding of the fluctuations controlling the physical behavior."
i) In how far is this a property of the SBE in particular, or of bosonic channel-decomposition more generally?
ii) In how far is this statement dependent on the bare model? E.g. what would happen with a J-term - see general comments?
* * *
page 15 - optional:
quote: "We remark that although the influence of the rest function on the s-wave susceptibilities remains small, its absolute values may not be small."
What is the physical interpretation of this? If a large rest function translates into 2PI-contributions becoming large, this might imply some physics, if I am not mistaken. (C.f. general comments on divergencies in 2PI terms)
* * *
page 17 - optional:
quote: "We note that the increase of the computational cost is significant and is reflected in the increased run times with respect to calculations where the filling is renormalized by the flow. "
-> Quantitative information would be informative here (c.f. general comments on computational matters).
* * *
page 17 - recommended:
quote: "This is surprising, since at finite doping d-wave fluctuations are expected to play an important role in the Hubbard model. However, these typically arise at lower temperatures."
I am a bit puzzled: If it is "surprising", then it calls for further investigation and verification. If it is not, due to elevated temperatures, then why make that very statement?
-> I suggest to clarify this matter. Ideally, if/since these fluctuations arise at lower temperatures, this case could simply be looked at using the very method of SBE-fRG (c.f. suggestions for additional results in the general comments).
* * *
page 20 - recommendation:
In Fig. 10 the inverse of the susceptibilities is plotted. Yet, the blue line representing the magnetic channel is hardly recognisable quantitatively. It is thus not clear how big the quantity becomes near U'=0, and if it even diverges.
-> Suggestion: Maybe plot the magnetic part separately, or use logarithmic scaling, or use a second scale on the (right-hand) y-axis.
* * *
page 22 - recommendation:
quote: "... it can be described by an RPA-like formula with respect to the polarization of the Hubbard model at U′= 0. Importantly, the latter does include vertex corrections ..."
What does "the latter" refer to precisely? I do not quite understand in which respect the inclusion of vertex correction is emphasised here.
* * *
page 26 - recommendation:
quote: "... that allows us to perform numerically feasible scans of the parameter space."
Given that not the whole parameter space was and can be addressed here, I find the formulation "scans of the parameter space" potentially misleading. Rather, results in this manuscript are given for a limited set of points in parameter space, representative for a certain region, while other regions in that space are not being investigated here. I feel that this would be a valid option to supplement this manuscript or for future studies.
* * *
Recommendation
Ask for major revision
Strengths
- clear description of many technical details regarding the extension of the SBE formulation of FRG
- fair discussion of numerical approximations
Weaknesses
- applicability to more general models difficult to assess
- no explanation of the diagrammatics used in Figs.1-3
- extremely lengthy results section
Report
In their manuscript, the authors apply the SBE formulation of the FRG to a square-lattice Hubbard model with on-site as well as nearest-neighbor interactions. They find that a cancellation of $pp$ and $\bar{ph}$ contributions allows for the density-channel RPA to be valid. While the manuscript is technically sound and clearly written, I doubt that their work "open[s] a new pathway in an existing or a new research direction, with clear potential for multi-pronged follow-up work":
1. The long-ranged SBE decomposition they present in this work has been used previously (they cite Refs. [70,74,75]). An extension within the scope of FRG seems natural.
2. The authors restrict themselves to an extremely simple model in a straightforward regime. This raises the following questions: What are the strengths and weaknesses of their methodology when it is applied to more general models? In particular, I am wondering how their approximation of diagonal formfactor indices in the loop integrals works for nontrivial models like the Kagome lattice (where bond fluctuations are the dominant ones in large areas of phase space; since the authors cited Refs. [50,52] they should be aware of this).
3. A follow-up to point 2 arises when thinking about the applicability to general models: What is the complexity of the flow equations? What is the (numerically) most demanding part? The precise information on their bosonic momentum and frequency meshes in the main text (Tab.1, Fig.4) does not make any sense without knowing how the implementation deals with the given mesh sizes.
4. The notation of the SBE equations taken in this manuscript does not allow for a swift extension to models with multiple orbitals per site, lack of $SU(2)$ symmetry, multiple sites per unit cell. Instead it is focused on the one-band Hubbard model. Taking these equations to more general models has been proven to be technically challenging in the past (see the developments of TUFRG).
Together, points 1-4 render this work a very specific study of the nearest-neighbor Hubbard model using the SBE formulation of FRG, lacking general applicability. I see this manuscript fit in a more specialized journal (maybe SciPost Physics Core). Irrespective of the choice of specific journal, the open source nature of SciPost demands making the simulation code public.
Some additional remarks are given in the "Requested changes" box below:
Requested changes
- The authors cite a very constrained selection of FRG works in the context of TUFRG and moire materials. Please have a look at the current advancements of applicability of TUFRG for materials simulations.
- On p. 7, the authors stick to the physical channels. Can a similar decomposition also be achieved in the diagrammatic channels? Would this decomposition be any simpler?
- Is it possible to employ FFTs for the $\boldsymbol{k}$-space part of the loop integrals? They could in principle speed up the calculations massively (I could have had a look at the code, but it's not publicly available).
- following the above request: make the code publicly available
- please make the results section more concise by, e.g., moving some information to the appendices
Recommendation
Accept in alternative Journal (see Report)
Aiman Al-Eryani on 2025-01-17 [id 5134]
The authors felt a necessity to promptly comment on the first report (Report #2), that primarily challenged the originality and general applicability of the formalism presented.
We are confused by the claim about originality as the Refs. [70,74] represent only a step in the direction of the SBE formalism, without developing the concept of interaction reducibility beyond the RPA or FLEX approximations, and Ref. [75] shows a very specific application to the dual fermion formalism without discussing the --> nonlocal <-- renormalization of the three-leg vertex, which we show to be of fundamental importance.
The SBE approach has already proven to be a very fruitful framework to design affordable and insightful approximations, both in the framework of fRG[13,54] or self-consistent approaches [85], see also Phys. Rev. Research 6, 043159 (2024), but its advantages are restricted to models with local interaction. The "natural" or straightforward generalisation of the SBE to non-local interactions is addressed in Appendix B, and is largely distinct from the one we describe in the main text. In this light, our decomposition based on $\mathcal{B}$-reducibility is original and retains the efficiency of the original SBE as has been demonstrated here.
Regarding the generality and the choice of the model, we emphasise that in our calculations, in contrast to TUfRG studies to materials that the reviewer has aluded to, we include full frequency dependencies in the vertices as well as the self-energy feedback. In this context, the inclusion of many bands remains to be computationally prohibitive.
The idea presented here is however quite general, and can, within existing algorithmic and computational capabilities, be used to (efficiently) tackle a whole new class of models of physical interest, such as the Hubbard Holstein model, with complete considerations of frequencies and the self-energy.
The authors will address the other points in detail and modify their manuscript accordingly once all reports are available.

---

## Editorial Decision

unknown